# Characterization of Gut Microbiota Profile in Lipedema: A Pilot Study

**DOI:** 10.3390/nu17243909

**Published:** 2025-12-13

**Authors:** Laura Di Renzo, Giulia Frank, Barbara Pala, Rossella Cianci, Gemma Lou De Santis, Francesco Nicoletti, Giulia Bigioni, Moreno Ortoman, Marina Borro, Maurizio Simmaco, Daniele Peluso, Antonino De Lorenzo, Paola Gualtieri

**Affiliations:** 1Section of Clinical Nutrition and Nutrigenomics, Department of Biomedicine and Prevention, University of Rome Tor Vergata, Via Montpellier 1, 00133 Rome, Italy; laura.di.renzo@uniroma2.it (L.D.R.); gemmaloudesantis@gmail.com (G.L.D.S.); franicoletti18@hotmail.it (F.N.); bigionigiulia@gmail.com (G.B.); moreno_ortonam@hotmail.com (M.O.); delorenzo@uniroma2.it (A.D.L.); paola.gualtieri@uniroma2.it (P.G.); 2PhD School of Applied Medical-Surgical Sciences, University of Rome Tor Vergata, Via Montpellier 1, 00133 Rome, Italy; giulia.frank@ymail.com (G.F.); barbara.pala93@gmail.com (B.P.); daniele.peluso@uniroma2.it (D.P.); 3School of Specialization in Food Science, University of Tor Vergata, Via Montpellier 1, 00133 Rome, Italy; 4Department of Translational Medicine and Surgery, Catholic University of the Sacred Heart, 00168 Rome, Italy; 5Istituto di Ricovero e Cura a Carattere Scientifico (IRCCS), Fondazione Policlinico Universitario A. Gemelli, 00168 Rome, Italy; 6Department of Neuroscience, Mental Health and Sense Organs NESMOS, Sapienza University of Rome, 00185 Rome, Italy; marina.borro@uniroma1.it (M.B.); maurizio.simmaco@uniroma1.it (M.S.)

**Keywords:** lipedema, gut microbiota, body composition, Predictive, preventive, personalized and participatory (4P) medicine

## Abstract

**Background**: Lipedema is a progressive disorder of subcutaneous connective tissue, predominantly affecting women, and characterized by an increase in subcutaneous adipose tissue, particularly in the lower body. This study aims to explore the gut microbiota (GM) profile in lipedema patients to characterize the associated GM and compare it with the control group. **Methods**: A prospective randomized case–control pilot study was conducted from September 2023 to May 2024, involving 55 Caucasian women, aged 20–60. The participants were divided into two groups: 35 with lipedema (LIPPY) and 20 controls (CTRL). Body composition was assessed using Dual X-ray Absorbimetry (DXA), and GM analysis was performed through 16S rRNA gene sequencing. **Results**: LIPPY subjects showed increased Intramuscular Adipose Tissue (IMAT) and reduced Lean Mass (LM)/Fat Mass (FM) ratios. While alpha and beta diversity metrics did not differ significantly between groups, differential abundance analysis identified a significant reduction in *Eggerthellaceae* (Log Fold Change (LFC) = −0.19, *p* = 0.04) and enrichment of *Propionibacteriaceae* (LFC = +0.18, *p* = 0.009) and *Acidaminococcaceae* (LFC = +0.32, *p* = 0.013) in the LIPPY group. Genus-level analysis showed a significant reduction in *Blautia* and *Ruminiclostridium* (LFC = −0.32 and −0.02; *p* = 0.02 and 0.04) and enrichment of *Anaerostipes*, *Propionibacterium*, and *Phascolarctobacterium* (LFC = +0.07, +0.17, and +0.34; *p* = 0.02, 0.005, 0.005, respectively). In correlation analyses, within LIPPY, *Eggerthellaceae* correlated negatively with Body Mass Index (BMI) (ρ = −0.61, *p* < 0.05) and positively with Appenicular (Appen) LM/Weight and AppenLM/BMI (ρ = +0.43 and +0.41, *p* < 0.05), while *Anaerostipes* correlated positively with these lean mass indices (ρ = +0.40, *p* < 0.05). In CTRL, only *Anaerostipes* showed a significant negative correlation with BMI (ρ = −0.64, *p* < 0.05). **Conclusions**: This study provides the first evidence of a distinct GM profile in LIPPY, with notable links to adverse body composition markers such as IMAT. Trial Registration: Trial registered on 24 June 2013 with ClinicalTrial.gov (NCT01890070).

## 1. Background

Lipedema is a chronic, progressive disorder of subcutaneous adipose tissue (SAT) that almost exclusively affects women. It is characterized by a symmetrical and disproportionate enlargement of the lower limbs and often the buttocks and, in some cases, the arms, with relative sparing of the hands and feet [1]. Clinically, affected areas show abnormal SAT deposition with nodularity and increased tissue fragility, accompanied by pain, tenderness to palpation, easy bruising and a feeling of heaviness, frequently associated with edema and reduced quality of life [2]. Lipedema must be distinguished from generalized obesity and from primary or secondary lymphedema; nevertheless, mixed phenotypes such as lipo-lymphedema may occur in more advanced stages, when chronic overload of the lymphatic system leads to overt lymphatic dysfunction [3,4].

From a pathophysiological standpoint, lipedema is considered a genetic, inflammatory, chronic-degenerative and disabling disorder of the subcutaneous connective tissue [1]. Its prevalence has been estimated at approximately 1 in 72,000 inhabitants, but this figure is likely underestimated because of frequent misdiagnosis with other conditions such as obesity, lymphedema, localized adiposity and cosmetic skin alterations [2].

Lipedema is driven by a genetic background influenced by hormonal mechanisms, which account for its higher incidence in women. Connective tissue homeostasis alterations also contribute to its pathogenesis. The familial occurrence of lipedema suggests a genetic etiology; however, the mode of inheritance is challenging to determine and has been hypothesized to be either X-linked dominant or autosomal dominant with sexual limitation [5]. Familial clustering and next-generation sequencing data support a polygenic model in which rare variants in multiple “susceptibility genes” involved in steroidogenesis, extracellular matrix organization, lymphangiogenesis, lipid homeostasis and insulin signaling contribute to disease risk [2,5,6,7]. Case-control studies have also reported an increased frequency of functional polymorphisms in genes related to methyl-group metabolism and inflammation, such as methylenetetrahydrofolate reductase (MTHFR) and interleukin-6 (IL-6), suggesting that impaired onecarbon metabolism and pro-inflammatory cytokine signalling may further modulate individual susceptibility [6,7]. Estrogen signalling dysregulation has been implicated as a key hormonal driver, favouring lower-body SAT expansion through depot-specific effects on adipocyte proliferation, differentiation and lipid storage [8]. In parallel, alterations in elastic tissue, abnormal vascularization and microcirculatory dysfunction have been proposed to underlie the characteristic SAT and lymphatic changes observed in lipedema [1]. Within this framework, gene-environment interactions, including those mediated by the gut microbiota, are likely to be relevant: host variants affecting estrogen and cytokine pathways may influence the intestinal immune-metabolic milieu and thus shape gut microbiota (GM) composition, while microbiota-derived metabolites can in turn impact systemic inflammatory and metabolic networks that are dysregulated in lipedema [9,10].

On this genetically and hormonally primed background, lipedema manifests clinically as a disorder characterized by progressive lower-body SAT expansion, microvascular dysfunction and chronic low-grade inflammation in the affected tissues. In line with this, lipedema is also known as “painful fat syndrome” because of its characteristic pain and tenderness, and is frequently associated with edema and systemic inflammatory features [3,4]. The course of the disease is strongly influenced by associated comorbidities such as depression, obesity and lymphedema [2].

There is a bidirectional relationship between adipose and lymphatic dysfunction in lipedema, particularly evident in advanced stages [11]. The accumulation of adipose tissue induces chronic inflammation, increasing vascular permeability and interstitial fluid volume. Inflammatory cytokines impair lymphatic transport by reducing lymphangiogenesis and altering lymphatic vessel contractility [11]. The accumulation of inflammatory cells in the interstitium further burdens the lymphatic system, reducing its function. Excess macromolecules (lipids and proteins) in the interstitium affect the adipocyte environment, triggering adipocyte proliferation, hypertrophy, and differentiation of adipocyte progenitors, creating a vicious cycle driven by inflammation [11].

Given the strong inflammatory and adipose remodeling component observed in lipedema, increasing attention has been directed toward biological systems capable of modulating immune–metabolic balance. Among these, the GM has emerged as a key regulatory axis influencing adipose tissue function, systemic inflammation, and energy homeostasis. GM is a peculiar biological factor that mediates neuroendocrine and immune functions involved in body weight regulation and metabolic diseases [9]. One role of GM is to protect and support the intestinal mucosa in a close symbiotic relationship with the host organism. This close relationship allows GM to contribute to physiological homeostasis, and abnormalities in GM contribute to the development of many diseases [10].

In metabolic diseases, including obesity, GM is altered in a way that increases the absorption of energy from the diet. The GM plays an important role in regulating fat storage by modulating the development of substrates for the synthesis of storable fat [12]. The GM produces metabolites absorbed into the systemic circulation that are likely to contribute to the development of obesity-related complications by increasing inflammatory tissue damage [13]. Recent research has highlighted the differential interaction of GM with distinct adipose depots. Visceral white adipose tissue has been shown to interact dynamically with gut-derived metabolites such as Short Chain Fatty Acids (SCFAs) and endotoxins, contributing to adipocyte hypertrophy, macrophage infiltration, and systemic low-grade inflammation [14]. In contrast, SAT, although historically considered metabolically inert, is now recognized to undergo pathological remodeling in specific conditions. Despite this, little is known about the interplay between GM and SAT, especially in disorders such as lipedema, where SAT is selectively and disproportionately expanded [14].

To date, no studies have investigated GM composition in women with lipedema or its potential role in modulating adipose dysfunction and systemic metabolic alterations in this population. Therefore, this study aimed to identify a characteristic GM profile in lipedema patients (LIPPY) with respect to the healthy control group (CTRL).

## 2. Methods

### 2.1. Study Design

A prospective randomized case-control pilot study was conducted at the Clinical Nutrition and Nutrigenomics Section of the Department of Biomedicine and Prevention of the Faculty of Medicine at Tor Vergata University. The subjects were consecutively recruited as part of a routine medical examination program from September 2023 to May 2024. Inclusion criteria were female sex; consent to the protocol. Exclusion criteria were: age <20 and >60 years; diagnosis of lymphedema; stage IV lipedema; drug or alcohol abuse; psychotic disorders; acute or chronic renal failure; hepatic failure; neoplastic diseases under treatment; pregnancy and lactation; hepatitis B and C; HIV/AIDS; COVID-19; and administration of antibiotics, prebiotics, probiotics, or synbiotics within the past 30 days.

Informed consent was obtained from all subjects following the principles of the Declaration of Helsinki.

All patients underwent investigations, according to the experimental protocol, approved by the Ethics Committee of the Calabria Region Center Area Section (Register Protocol No. 97, 20 April 2023), to perform a differential analysis of the GM in LIPPY compared with the CTRL. No circulating inflammatory, hormonal, or muscle-related biomarkers were assessed as part of this study protocol; our work focused on GM analysis (16S rRNA sequencing) and DXA-based body composition assessment. The study design flow chart is reported in Figure 1.

### 2.2. Body Composition Evaluation

After a 12 h overnight fast, all subjects underwent anthropometric assessment (body weight, height, waist, and hip circumference) according to the standard method [15]. All individuals were instructed to remove their clothes (except underwear), shoes, and any metal objects before being measured. BMI was calculated using the following formula: BMI = body weight (kg)/height (m)^2^. Body composition was assessed according to the standard method [16]. Whole and segmental Fat Mass (FM) (kg) was assessed by Dual X-ray Absorbimetry (DXA) (Primus, X-ray densitometer; software version 1.2.2, Osteosys Co., Ltd., Guro-gu, Seoul, Republic of Korea) [17]. Total FM (% FM) was calculated as total body FM (total FM) divided by the total mass of all tissues, including total body bone (TBBone) and Lean Mass (LM), as follows:% FM = [Total FM/(Total FM + Total LM + TBBone)] × 100

Women were classified as LIPPY according to clinical criteria derived from current lipedema diagnostic guidelines and expert consensus statements [18]. Diagnosis required a bilateral and largely symmetrical, disproportionate increase in SAT of the lower limbs (with or without involvement of hips, buttocks and, in some cases, arms), with relative sparing of the feet and hands and, when present, a characteristic ankle or wrist “cuff”. On physical examination, the affected tissue showed a soft-to-nodular or “pebbly” consistency, sometimes with palpable larger nodules in more advanced presentations, and was associated with pain or tenderness to palpation, easy bruising and a subjective feeling of heaviness. All women classified as LIPPY reported chronic and progressive symptoms, with onset or clear worsening during periods of hormonal transition (puberty, pregnancy or menopause), and most indicated a family history suggestive of familial aggregation of the disease. In terms of distribution and morphology, the majority of cases were consistent with type II-III involvement (from below the umbilicus to the knees or ankles), with some patients also showing type IV arm involvement and stage 1-3 skin changes, as described in published staging and typing systems. Participants with clinical signs of primary or secondary lymphedema (e.g., positive Stemmer sign, marked pitting edema, unilateral or clearly asymmetric swelling) or with isolated localized adiposity without pain or typical tissue changes were not classified as lipedema.

### 2.3. Gut Microbiota Analysis

Each woman received a standardized fecal collection kit accompanied by detailed written and verbal instructions to ensure proper handling and collection of stool samples. Participants collected a single fecal sample at home, immediately after defecation, carefully avoiding contamination with urine or water. Samples were returned to the clinic according to the instructions provided and then shipped under controlled conditions to Wellmicro^®^ (Via Antonio Canova 30, 40138 Bologna, Italy) for microbiota analysis [19].

Bacterial DNA was extracted from stool using a standardized mechanical and chemical lysis protocol followed by automated purification, according to Wellmicro’s validated procedures [19]. The hypervariable V3-V4 regions of the bacterial 16S rRNA gene were amplified and sequenced on an Illumina MiSeq platform using a next-generation sequencing (NGS) workflow. Raw sequences underwent bioinformatic processing with QIIME2 (v.2021.11) and DADA2 (v.1.16) for quality filtering, denoising, chimera removal, and inference of amplicon sequence variants (ASVs). Taxonomic assignment was performed using a proprietary bacterial reference database in which the main public databases (Greengenes, SILVA, and NCBI) are merged and harmonized. Relative abundances were obtained at phylum, family, and genus levels and used for downstream diversity and differential abundance analyses as described in the Statistical Analysis section.

### 2.4. Statistical Analysis

Data were collected on an Excel spreadsheet (2020, Microsoft, Redmond, WA, USA). Statistical analyses were performed using R (version 4.4.2). Descriptive statistics were conducted for demographic and anthropometric characteristics of patients. Continuous variables were summarized using the mean and standard deviation (SD), while categorical variables were presented as frequency and percentage.

A Shapiro-Wilk test was performed to assess the normality of the sample distribution. Based on this, we employed a nonparametric Wilcoxon test to assess differences in body composition parameters between CTRL and LIPPY groups. Spearman’s correlation (ρ) was calculated for all numerical variables, with Benjamini–Hochberg multi-test correction.

Microbial differential abundance analysis between groups was performed using the ANCOM-BC method. This approach accounts for compositional data structure and corrects for sampling biases. Abundance data were normalized, and taxa with adjusted *p*-values < 0.05 were considered significantly different. Results are reported as log fold changes (LFCs) between LIPPY and CTRL groups. To compare differences in microbial community structure between the two groups, both alpha and beta diversity metrics were evaluated. Alpha diversity was assessed using the Shannon index, which captures both richness and evenness of microbial communities and is widely used in 16S rRNA gene sequencing studies because it is relatively robust to differences in sequencing depth and appropriate for compositional data. Other richness estimators, such as Observed richness or Chao1, which are more sensitive to sequencing depth and rare taxa detection, were therefore not selected. Beta diversity was computed using Bray-Curtis dissimilarity and visualized through Principal Coordinates Analysis (PCoA) plots; statistical significance of group-level clustering was tested using Permutational Multivariate Analysis of Variance (PERMANOVA).

Statistical significance was evaluated using Satterthwaite’s approximation for degrees of freedom to compute *p*-values. For beta diversity, differences in microbial community composition between diet groups were assessed using PERMANOVA based on Bray-Curtis dissimilarity matrices.

A *p* value of <0.05 was considered statistically significant.

## 3. Results

### 3.1. Study Population

The study included 80 participants and assigned them to two study groups: LIPPY and CTRL. In total, 25 patients were excluded from the study because 18 were given antibiotics, 3 became pregnant, and 4 did not sign the informed consent. The study included a total of 55 Caucasian women, aged between 20 and 60 years, all recruited from the Italian population. Among them, 35 were diagnosed with LIPPY, while the remaining 20 participants formed the CTRL group.

The two groups were comparable in terms of height. The average height in the overall sample was 161.72 cm (±5.92)**,** with no meaningful difference between the LIPPY group (161.74 cm ± 5.45) and the CTRL group (161.67 cm ± 7.30, *p* = 0.92). The mean body weight was 72.17 kg (±5.37) in the LIPPY group compared to 68.85 kg (±6.01) in the CTRL group (*p* = 0.06).

Notably, the LIPPY group showed significant differences across multiple anthropometric and body composition variables when compared to the CTRL group. Specifically, BMI was 63.00% higher (34.50 ± 10.70 vs. 21.20 ± 2.80 kg/m^2^, *p* < 0.01), IMAT was elevated by 70.00% (1.07 ± 0.46 vs. 0.63 ± 0.25 kg, *p* < 0.01), and the LM/FM Trunk was 44.00% lower (1.58 ± 0.94 vs. 2.83 ± 1.21, *p* < 0.01). AppenLM adjusted for both weight and BMI was also statistically significant (0.24 ± 0.07 vs. 0.29 ± 0.03, and 0.62 ± 0.18 vs. 0.77 ± 0.13; *p* < 0.01 and *p* < 0.01, respectively).

All variables with *p* < 0.05 are summarized in Table 1.

### 3.2. Microbiome Composition and Diversity

Microbial profiling detected 272 operational taxonomic units (OTUs). Approximately 67.00% were shared between groups, while 21.20% were exclusive to CTRL and 12.00% to LIPPY.

The top 10 most abundant phyla, families, and genera in both groups are shown in Figure 2.

Alpha diversity metrics, including Shannon and Inverse Simpson indices, did not differ significantly between groups (*p* > 0.05) (Figure 3).

Similarly, beta diversity, analyzed via Bray–Curtis and Jaccard distances in PCoA plots, showed no significant group separation (Figure 4). PERMANOVA confirmed the absence of global GM composition differences (R^2^ = 0.02, *p* = 0.35), and homogeneity of dispersions was verified (permutation test, *p* = 0.71).

### 3.3. Differential Abundance

Using the ANCOM-BC algorithm, differential abundance analysis revealed three bacterial families with statistically significant differences between the LIPPY and CTRL groups (*p* < 0.05). Among them, *Eggerthellaceae* was significantly decreased in the LIPPY group (LFC = −0.19, *p* = 0.04). In contrast, *Propionibacteriaceae* and *Acidaminococcaceae* were significantly enriched in the LIPPY group, with positive log fold changes of 0.18 (*p* = 0.009) and 0.32 (*p* = 0.01), respectively. Moreover, statistically significant differences at the genus level were observed. In particular, *Blautia* and *Ruminiclostridium* were significantly decreased in the LIPPY group compared to CTRL (LFC = −0.32 and −0.02; *p* = 0.02 and 0.04, respectively), whereas *Anaerostipes*, *Propionibacterium*, and *Phascolarctobacterium* were significantly enriched (LFC = +0.07, +0.17, and +0.34; *p* = 0.02, 0.005, and 0.005, respectively). Results are shown in Figure 5 and Figure 6.

### 3.4. Correlations Between Microbiota and Body Composition

Correlation matrices between discriminative microbial taxa and body composition parameters revealed meaningful patterns. In the LIPPY group (Figure 7a), *Eggerthellaceae* were negatively correlated with BMI (ρ = −0.61, *p* = 0.0003) and positively with AppenLM/weight and AppenLM/BMI (ρ = +0.43 and +0.41, *p* = 0.01 and 0.02, respectively).

Moreover, in this group, *Anaerostipes* were positively correlated with AppenLM/weight and AppenLM/BMI (ρ = −0.43 and −0.37, *p* = 0.01 and 0.04, respectively).

Interestingly, in the CTRL group (Figure 7b), only *Anaerostipes* were negatively associated with BMI (ρ = −0.64, *p* = 0.02).

## 4. Discussion

This pilot study provides preliminary evidence of distinct GM and body composition features in LIPPY compared to CTRL. Taxonomic analysis revealed a reduction in *Eggerthellaceae* and enrichment of *Propionibacteriaceae* and *Acidaminococcaceae* in LIPPY. These findings were paralleled by alterations in body composition, particularly a significant increase in IMAT, and a lower LM/FM of the trunk, AppenLM/Weight, and AppenLM/BMI, as assessed via DXA.

These findings challenge earlier characterizations of lipedema as a “metabolically benign” adipose tissue disorder. Prior DXA-based studies documented disproportionate fat accumulation in the lower limbs with preservation of lean mass and no evidence of ectopic fat deposition [20,21,22]. In contrast, our observation of increased IMAT, a recognized ectopic fat depot, suggests a less favorable metabolic phenotype in LIPPY.

This aligns with emerging evidence indicating that lipedema may involve broader metabolic dysregulation, particularly when muscle fat infiltration is present. IMAT accumulation is increasingly viewed not as a passive storage depot but as a biologically active compartment associated with a pro-inflammatory microenvironment, mitochondrial dysfunction, and impaired myocyte insulin signaling. Mechanistically, fatty infiltration in skeletal muscle has been shown to disrupt insulin receptor substrate-phosphatidylinositol 3-kinase-Akt (IRS-PI3K-Akt) pathways and to reduce glucose transporter type 4 (GLUT4) translocation, thereby contributing to local insulin resistance and reduced glucose uptake [23,24,25]. In parallel, adipocytes within the muscle milieu secrete pro-inflammatory cytokines such as TNF-α and IL-6, which exacerbate metabolic stress, alter mitochondrial function and interfere with muscle repair and contractile properties [26]. Moreover, elevated IMAT has been linked to reduced capillary density, impaired oxidative capacity, and increased muscle stiffness, features also observed in aging, obesity, and sarcopenia [27]. In the context of lipedema, such infiltration may further contribute to the progressive loss of mobility, disproportionate fatigue, and reduced exercise tolerance frequently reported by patients [28,29]. Thus, the presence of IMAT not only reflects altered fat distribution but may also participate in the metabolic and functional impairment observed in more advanced stages of the disease, although this remains to be confirmed in longitudinal studies.

Although mean height and weight appeared similar between groups, the LIPPY cohort exhibited a markedly higher intra-group variability in body weight, with several individuals presenting disproportionately elevated values. This wider dispersion is consistent with the well-recognized heterogeneity of body composition in lipedema, where excess adipose tissue is preferentially distributed in the lower limbs and trunk despite normal or moderately increased total body weight. Previous reports have similarly shown that women with lipedema may present normal or only mildly elevated body weight but disproportionately high BMI, as BMI does not account for abnormal subcutaneous fat accumulation that is characteristic of the disease [30]. Given the comparable height between groups, this variability in weight translated into significantly higher BMI values in the LIPPY group, reflecting the disproportionate FM characteristic of lipedema rather than generalized obesity. These observations align with prior studies reporting discordance between BMI and classical obesity markers in lipedema, further supporting the inadequacy of BMI as a standalone metric for characterizing adiposity phenotypes in this condition.

In addition, the combination of reduced AppenLM relative to weight or BMI and increased IMAT in LIPPY is compatible with body-composition profiles that, in other populations, have been associated with impaired physical performance and increased risk of mobility limitation. Indices of skeletal muscle mass adjusted for adiposity, such as lean mass-to-BMI or skeletal muscle mass relative to body fat, correlate more strongly with gait speed and incident disability than absolute lean mass alone [31]. Likewise, greater intermuscular or intramuscular adipose tissue has been repeatedly linked to slower walking speed, reduced muscle strength, and poorer lower-extremity function in older adults and in individuals with metabolic disease [32]. Although our study did not include direct functional assessments, these converging data suggest that the IMAT-rich, low-lean-mass pattern observed in LIPPY may represent an early functional vulnerability. To substantiate the hypothesis that IMAT represents a metabolically adverse phenotype in lipedema and a potential marker of disease progression, longitudinal and mechanistic studies will be required. Future research should combine repeated body-composition imaging (e.g., DXA or magnetic resonance imaging), histological characterization of muscle tissue and the assessment of early circulating biomarkers of metabolic stress and muscle quality, in order to validate IMAT as a biomarker of functional decline. In parallel, recent evidence has highlighted musculoskeletal ultrasound as a reliable tool for the evaluation of superficial soft tissues and lymphatic alterations in conditions such as lymphedema, providing dynamic, bedside information on tissue architecture and fluid accumulation [33,34]. The development of standardized ultrasound protocols tailored to lipedema could facilitate more consistent assessment of subcutaneous and, where feasible, intermuscular fat compartments, including IMAT, and help integrate these measures into future diagnostic and monitoring frameworks.

From a gut microbiota perspective, alpha diversity metrics (including Shannon and Inverse Simpson indices) did not differ significantly between LIPPY and CTRL. The lack of significant differences in alpha diversity indicates that lipedema does not appear to reduce overall microbial richness or evenness. These indices reflect both the number of taxa and their relative distribution within the community, and their similarity between groups suggests that the microbiota of women with lipedema is not globally reduced or simplified. Rather, our findings are more consistent with a compositional rearrangement of specific taxa related to lipedema, instead of a generalized loss of bacterial diversity.

The microbial alterations observed in our study are consistent with a potential involvement of GM in lipedema. The enrichment of *Propionibacteriaceae* and *Acidaminococcaceae*, families involved in amino acid fermentation and SCFAs production, particularly acetate and propionate, suggests a modification of microbial metabolic output in LIPPY [35]. SCFAs are known to modulate gut hormone release (GLP-1, PYY), regulate intestinal barrier integrity, and impact insulin sensitivity. However, the direction and clinical relevance of these changes in lipedema should be interpreted as hypothesis-generating, especially in a metabolically heterogeneous setting.

Among the taxa differentiating the two groups, *Eggerthellaceae* emerged as clinically and biologically plausible candidates of interest. This family includes genera such as *Slackia* and *Gordonibacter*, which are involved in the metabolism of bile acids, steroid hormones, and dietary polyphenols, forming a key component of the intestinal “estrobolome” [36]. Through bacterial β-glucuronidase and β-glucosidase activity, the estrobolome modulates enterohepatic estrogen recycling, and thereby contributes to systemic estrogen exposure and downstream inflammatory and metabolic pathways [37,38]. Experimental and clinical data indicate that alterations in estrogen receptor pathways can promote adipocyte hypertrophy, shift fat distribution toward more metabolically adverse depots and enhance pro-inflammatory signalling within white adipose tissue [39]. Given that lipedema is a hormonally sensitive disorder, typically manifesting or worsening during puberty, pregnancy, or menopause [8], a reduction in estrogen-metabolizing taxa such as *Eggerthellaceae* could, in principle, disturb estrogen homeostasis and favour estrogen-dependent adipose expansion and low-grade inflammation in affected depots. However, these links remain hypothetical in the absence of direct measurements of estrogen metabolites and tissue inflammatory markers in lipedema.

In our study, *Eggerthellaceae* were significantly reduced in LIPPY compared with CTRL and showed a distinct pattern of association with body composition parameters. Within the LIPPY group, higher *Eggerthellaceae* abundances correlated negatively with BMI (ρ = −0.61, *p* < 0.05) and positively with appendicular lean-mass indices (AppenLM/Weight: ρ = +0.43; AppenLM/BMI: ρ = +0.41; both *p* < 0.05), suggesting that in this specific context, greater *Eggerthellaceae* levels are observed in women with relatively lower adiposity and better preserved muscle mass. This finding is consistent with the notion that microbes involved in estrogen and bile acid metabolism may be linked to inflammatory control, energy balance and lipid partitioning [40], but causal relationships cannot be inferred from cross-sectional associations.

Genus-level analysis further confirmed significant compositional differences in LIPPY. *Blautia* and *Ruminiclostridium* were significantly decreased (LFC = −0.32 and −0.02; *p* = 0.02 and 0.04, respectively), while *Anaerostipes, Propionibacterium,* and *Phascolarctobacterium* were significantly enriched (LFC = +0.07, +0.17, and +0.34; *p* = 0.02, 0.005, and 0.005, respectively). A significant reduction in *Blautia* was observed in LIPPY. This genus has been repeatedly associated with improved insulin sensitivity and anti-inflammatory properties, and several species are known producers of beneficial metabolites linked to better glucose homeostasis, hepatic lipid metabolism and SCFA-mediated improvements in insulin signalling. In particular, *Blautia* has been proposed as a functional genus with potential probiotic properties In metabolic and inflammatory conditions [41,42,43]. Therefore, its reduction in our LIPPY cohort may reflect a shift towards a less favourable metabolic gut environment, in line with observations reported in metabolic syndrome and obesity. This finding supports the emerging view that lipedema may share some microbiota-related mechanisms with metabolic disorders, although this interpretation remains exploratory in the absence of mechanistic data.

*Ruminiclostridium*, a genus within the *Ruminococcaceae* family, includes short-chain fatty acid–producing species in the colon and has been linked to obesity and cardiometabolic traits in humans, for example, *Ruminiclostridium 5* has been associated with SCFA production and cardiometabolic parameters in children [44]. However, functional data on this genus in the context of lipedema are lacking, and the clinical relevance of its reduced abundance in LIPPY remains uncertain. *Propionibacterium*, a propionate-producing genus, contributes to gluconeogenesis and satiety signaling, but its increased abundance has also been linked to adiposity and metabolic inflammation in dysbiotic conditions [45].

*Phascolarctobacterium*, another propionate producer, was enriched in LIPPY. Previous work has associated higher *Phascolarctobacterium* abundance with better physical function, including improved gait speed and cardiorespiratory fitness, in adults with chronic HIV infection, and a recent review has highlighted positive associations with muscle mass maintenance, mitochondrial efficiency and prevention of functional decline [46,47]. In our study, *Phascolarctobacterium* showed positive correlations with lean mass indices in women with lipedema, which is compatible with these earlier observations. However, given the cross-sectional design and the small sample size, these parallels should be interpreted with caution and regarded as hypothesis-generating.

*Anaerostipes* deserves particular attention. This taxon was significantly enriched in LIPPY and showed a distinct pattern of association with body composition parameters. *Anaerostipes* is generally regarded as a beneficial butyrate-producing genus [48], and butyrate has been shown to promote muscle protein synthesis, mitochondrial biogenesis, and satellite cell differentiation, thereby supporting muscle mass and function [49]. In our cohort, *Anaerostipes* correlated positively with appendicular lean mass indices (AppenLM/Weight and AppenLM/BMI) within the LIPPY group (ρ ≈ +0.40, *p* < 0.05), suggesting a potential role in preserving or enhancing skeletal muscle mass in the specific metabolic and inflammatory context of lipedema. In contrast, in the CTRL group, the only significant association was an inverse correlation between *Anaerostipes* and BMI (ρ = −0.64, *p* < 0.05), in line with previous evidence linking butyrate-producing bacteria to lower adiposity and improved metabolic profiles [50]. The absence of lean mass correlations in healthy controls, coupled with the positive associations observed in LIPPY, suggests that the relationship between *Anaerostipes* and body composition may be context-dependent. Whether this reflects a compensatory response or a disease-related remodeling of the gut–muscle–fat axis cannot be determined from the present data.

Taken together, the positive associations of SCFA-producing genera such as *Anaerostipes* and *Phascolarctobacterium* with lean mass indices in LIPPY are compatible with the emerging concept of a muscle–adipose–microbiota axis. SCFA-producing taxa have been implicated in the regulation of mitochondrial function, muscle protein synthesis and inflammatory signalling [46,51,52,53]. In our cohort, these taxa co-occurred with a more favourable appendicular lean mass profile, but the cross-sectional nature of the study and the small sample size do not allow us to establish directionality or causality. These links should therefore be regarded as hypothesis-generating and need to be tested in mechanistic and interventional studies.

When viewed within the broader context of metabolic and inflammatory disease models, the microbial profile observed in LIPPY shows partial overlap with patterns described in obesity and metabolic syndrome. Combinations of lower *Blautia* abundance, alterations in SCFA-producing and propionate-producing genera, and reduced *Eggerthellaceae* have been reported in association with chronic low-grade inflammation, altered lipid handling, and cardiometabolic risk [43,54,55]. Taken together, these parallels suggest that part of the microbiota signature identified in lipedema overlaps with profiles seen in other cardiometabolic conditions, raising the possibility that LIPPY could share some microbiota-related mechanisms of metabolic and cardiovascular vulnerability.

Beyond this general framework, the available genetic data help to mechanistically link lipedema-associated polymorphisms, adipose tissue inflammation and the GM profile described in our cohort. The multi-gene next-generation sequencing panel developed by Michelini et al. [5] identified rare variants in several genes that regulate lipid storage and lipolysis (e.g., PLIN1, LIPE), insulin and growth hormone signalling (INSR, GHR), nuclear receptors integrating fatty-acid- and bile-acid-derived signals (PPARG, PPARA, NR0B2) and intracellular cholesterol and glucose handling (NPC1, GCKR), supporting a polygenic background that favours adipocyte hypertrophy, ectopic fat deposition and metabolic inflexibility in susceptible depots. Case-control studies in Italian women with lipedema further show an over-representation of functional polymorphisms in IL-6 and in the C677T variant of MTHFR, which are associated with enhanced IL-6 production, hyperhomocysteinaemia and reduced methylation capacity, and may therefore promote endothelial dysfunction, microangiopathy and a pro-inflammatory cytokine milieu within lower-body SAT [6,7]. Histological and transcriptomic analyses of lipedema SAT confirm adipocyte hypertrophy, interstitial fibrosis, microvascular alterations and a predominance of CD68+/CD163+ macrophages, together with dysregulation of transcription factors such as CEBPD and KLF4 and of genes involved in adipogenesis, extracellular matrix remodelling and myeloid cell activation (e.g., CSF1R, TREM2, MAFB, C1Q, C2, CD84, CD209), delineating a typical picture of “sick fat” that differs from both simple obesity and lymphedema [56,57]. On this genetically determined background, inter-individual variation in adipogenic, inflammatory and vascular pathways is expected to modulate the intestinal immune-metabolic niche, via changes in adipokine and cytokine output, estrogen and bile-acid homeostasis and endothelial function, and thereby to shape GM composition [5,9,10,12]. In line with this interpretation, evidence from complex immune-metabolic diseases indicates that host variants in inflammatory and barrier-related pathways can constrain the range of microbiome configurations that are stably compatible with a given genetic background: in inflammatory bowel disease, genome-microbiome association studies link risk alleles in NOD2, TNFSF15 and IL12B, enriched in innate and JAK-STAT signalling, to characteristic taxonomic shifts and higher dysbiosis indices [58]. In obesity and insulin resistance, dysbiosis with increased intestinal permeability has been proposed to drive “metabolic endotoxemia”, whereby gut-derived lipopolysaccharides activate Toll-like receptor-4 and NF-κB, increase IL-6, TNF-α, IL-1β and hepatic C-reactive protein, disrupt tight junction proteins such as ZO-1 and promote low-grade inflammation and adipose-tissue insulin resistance; concomitantly, reduced short-chain fatty acid production and altered bile-acid and branched-chain amino-acid metabolism may reinforce adipocyte dysfunction and ectopic fat storage [52,53,59]. By analogy, in lipedema, a SAT phenotype genetically primed to overexpress IL-6 and to develop microangiopathy and adiposopathy may favour, and be particularly vulnerable to, a pro-inflammatory, endotoxemia-prone GM configuration. The GM signature observed in our LIPPY cohort, enriched in amino-acid- and propionate-producing taxa and reduced in estrogen-metabolizing and SCFA-producing microbes, appears to partially overlap microbiota patterns reported in obesity and metabolic syndrome (lower *Blautia* and *Eggerthellaceae* with enrichment of *Propionibacteriaceae*, *Acidaminococcaceae*, *Propionibacterium* and *Phascolarctobacterium*), but in our sample, these shifts correlated more strongly with increased IMAT and reduced lean mass-to-fat mass ratios than with generalized adiposity [51,54,55]. In this context, such a configuration may contribute to amplifying adipose and lymphatic inflammation via microbial metabolites (e.g., SCFAs, bile-acid derivatives, endotoxins), thereby sustaining SAT remodelling, IMAT accumulation and impairment of the muscle-fat axis [9,10,35]. Overall, these converging findings are consistent with a working model in which lipedema-associated polymorphisms predispose to a compartmentalized, estrogen-sensitive adiposopathy that interacts bidirectionally with the GM, establishing a vicious circle between lower-body SAT inflammation, dysbiosis and functional body-composition alterations.

It is important to acknowledge that these correlations are associative and may be influenced by unmeasured confounding factors. Key determinants of both GM composition and body composition, including habitual dietary intake, physical activity levels, and hormonal status, were not quantitatively assessed in this pilot study. Variations in fiber consumption, exercise volume, or estrogen exposure can independently shape microbial metabolic profiles (e.g., SCFA production) and modulate skeletal muscle and adipose tissue physiology. Therefore, these lifestyle- and hormone-related factors may partly contribute to the microbiota-lean mass associations observed here, and should be considered in future mechanistic and longitudinal studies.

Among these potential determinants, dietary patterns are particularly relevant, as they are major drivers of GM composition and function and may therefore be of interest in lipedema. High-fiber, plant-rich dietary models such as the Mediterranean diet and plant-based diets have been shown to promote SCFA-producing and anti-inflammatory taxa, increase overall microbial diversity, and improve gut barrier integrity [60]. In contrast, low-carbohydrate or ketogenic diets induce distinct shifts in GM structure and bile acid metabolism, with heterogeneous effects on metabolic and inflammatory pathways [61]. Beyond macronutrient patterns, bioactive compounds with antimicrobial and immunomodulatory properties, such as capsaicin and piperine, have been shown in experimental models to reshape gut microbiota, reduce endotoxemia and low-grade inflammation, and attenuate diet-induced obesity [62]. These observations raise the hypothesis that targeted dietary strategies, including Mediterranean-style or plant-forward patterns and the controlled use of specific phytochemicals, could be explored as adjunctive approaches to modulate GM and potentially ameliorate metabolic and inflammatory features in lipedema. However, these possibilities remain speculative at present, and the use of antibiotics solely to manipulate GM cannot be recommended in a chronic condition such as lipedema, given their profound and often detrimental impact on microbial ecology. Controlled dietary intervention studies are needed to clarify whether microbiota-directed nutrition can beneficially influence disease phenotype.

At this stage, the GM signatures identified in lipedema should be regarded as exploratory biomarkers that may help design future microbiota-directed interventions rather than as therapeutic targets per se. In principle, these patterns could be used to stratify participants and define mechanistic endpoints in trials testing Mediterranean-style or fibre- and polyphenol-rich dietary patterns, as well as selected pre-, pro- or post-biotic formulations, which in other metabolic settings have been shown to favour SCFA-producing and anti-inflammatory taxa and to improve host metabolic profiles.

Although our population encompassed women aged 20-60 years, the observed microbiota-muscle-adipose associations might also have translational implications for older adults. In populations in which adherence to both adequate protein intake and regular physical activity is challenging, microbiota-targeted strategies (for example, pre-, pro- or post-biotics) have been proposed as potential adjuncts to support muscle health by modulating SCFA-producing taxa, as suggested by studies in older adults and individuals with sarcopenic obesity and chronic disease [63,64]. In our cohort, SCFA-associated genera such as *Anaerostipes* and *Phascolarctobacterium* showed positive correlations with lean mass indices, which is consistent with this hypothesis. However, dedicated intervention studies in older adults with lipedema or related phenotypes would be required to test whether microbiota-based approaches can effectively contribute to muscle mass preservation in real-life settings.

Taken together, these genus-level and correlation-based findings indicate that GM in lipedema differs from that of healthy controls and may be functionally relevant to adipose and muscle phenotypes. However, given the small sample size and cross-sectional design, the results should be regarded as preliminary and hypothesis-generating, and further studies incorporating multi-omic and interventional approaches are needed to clarify causality and clinical implications.

Figure 8 reports a summary illustration of the main findings of the study.

### 4.1. Strenghts

This study has several noteworthy strengths. To our knowledge, it is the first investigation to explore GM composition in women with lipedema in conjunction with detailed body composition profiling, particularly focusing on muscle-adipose tissue relationships. Integrating microbiome analysis with high-resolution DXA provides novel insight into the microbial correlates of ectopic fat deposition and lean mass indices. Additionally, stringent inclusion and exclusion criteria, excluding recent antibiotic use, ongoing pharmacological therapies, and metabolic comorbidities, helped minimize potential confounders that frequently complicate GM-related studies.

### 4.2. Limitations

Nonetheless, certain limitations must be acknowledged. First, the cross-sectional design precludes causal inference, and the observed associations should be interpreted as hypothesis-generating. Second, although adequate for exploratory analyses, the relatively small sample size may limit statistical power, particularly for detecting subtle GM-phenotype interactions. Third, dietary intake, physical activity, and hormonal status, important determinants of both gut microbiota composition and body composition, were not quantitatively assessed, potentially introducing unmeasured confounding and partly contributing to the observed associations. A further limitation is that no circulating biomarkers (e.g., inflammatory cytokines or hormonal regulators of muscle and adipose tissue) were assessed, as the study was designed to focus exclusively on GM composition and DXA-derived body composition. Finally, while 16S rRNA sequencing allowed robust taxonomic profiling, it does not provide strain-level resolution nor direct information on microbial functional potential or metabolic output. As a result, we cannot define which specific metabolic pathways (e.g., short-chain fatty acid, bile acid or xenobiotic metabolism) underlie the observed taxonomic shifts, nor whether these changes entail predominantly beneficial, neutral or potentially adverse effects for the host. Future studies should therefore integrate shotgun metagenomics and targeted or untargeted metabolomics with longitudinal clinical follow-up, in order to characterise microbial gene content and circulating or faecal metabolites and thereby clarify the mechanistic, therapeutic and safety implications of microbiota alterations in lipedema.

## 5. Conclusions

This study provides the first preliminary evidence of a distinct GM signature in LIPPY, accompanied by unfavorable body composition features, including elevated IMAT and reduced AppenLM/weight and AppenLM/BMI. While global microbial diversity metrics were comparable between groups, taxonomic profiling revealed specific compositional shifts, notably, a reduction in *Eggerthellaceae* and enrichment of *Propionibacteriaceae* and *Acidaminococcaceae*, that may be relevant to disease pathophysiology.

Importantly, correlation analyses uncovered differential associations between microbial taxa and body composition metrics in lipedema, suggesting that GM may contribute to, or reflect, alterations in the muscle.adipose axis. The positive associations of *Eggerthellaceae* and *Anaerostipes* with AppenLM indices, together with their known roles in estrogen metabolism and butyrate production, point toward a potential protective or compensatory function in the context of adipose infiltration and altered hormonal signaling.

Although limited by its cross-sectional design, modest sample size, and lack of functional microbial data, this pilot study supports the hypothesis that GM may act as both a biomarker and a mechanistic player in lipedema. These findings underscore the need for future longitudinal and multi-omic studies to elucidate better the causal pathways involved and to evaluate GM-targeted interventions as potential strategies for improving metabolic and functional outcomes in this under-recognized condition.

## Figures and Tables

**Figure 1 nutrients-17-03909-f001:**
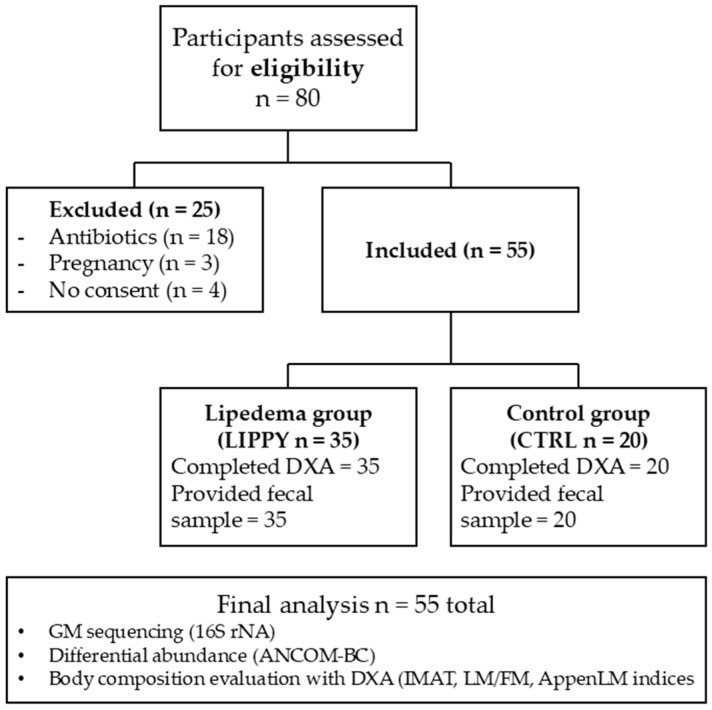
Study design flowchart.

**Figure 2 nutrients-17-03909-f002:**
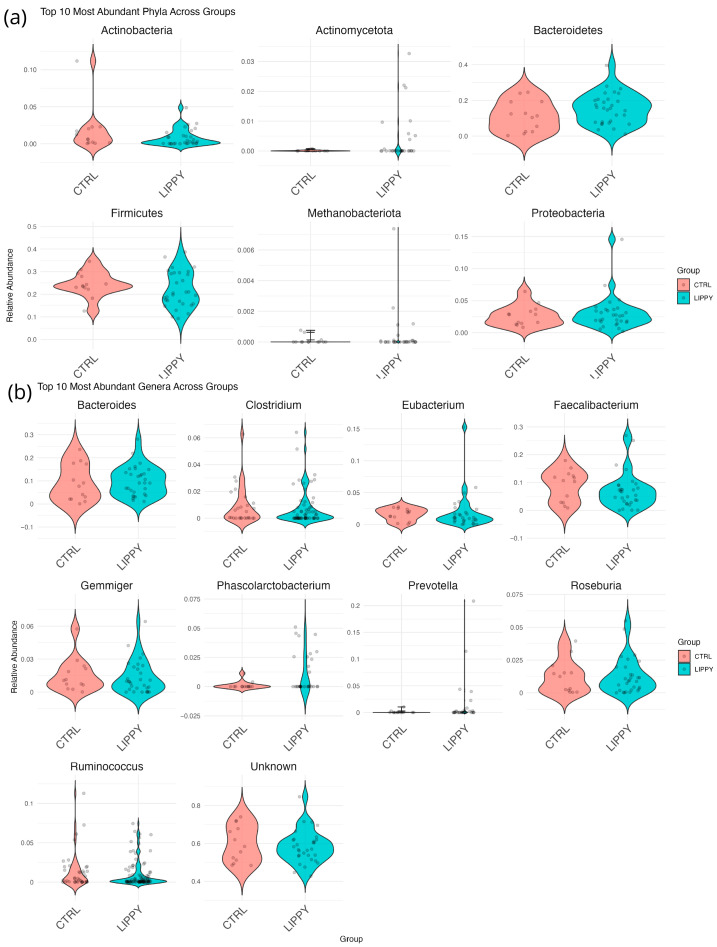
Microbial composition of LIPPY and CTRL groups. (**a**) Top 9 most abundant phyla, (**b**) Top 10 families, (**c**) Top 10 genera. Bar plots show mean relative abundances in each group.

**Figure 3 nutrients-17-03909-f003:**
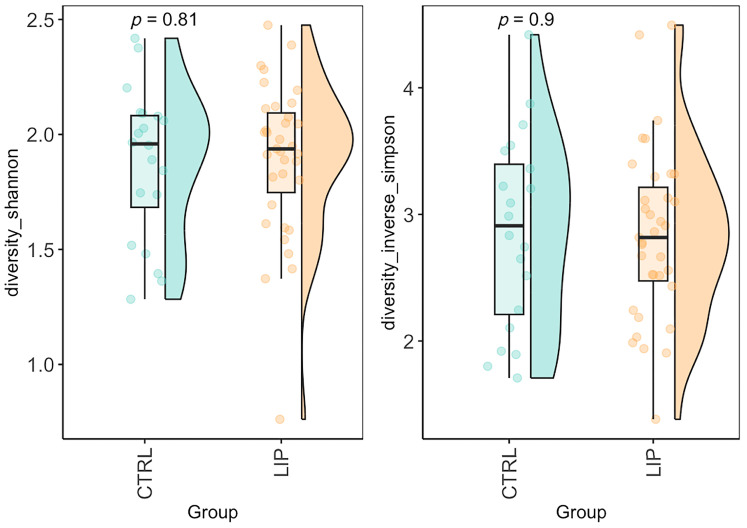
Alpha diversity of gut microbiota in control (CTRL) and Lipoedema (LIPPY) groups. Violin plots with overlaid boxplots and individual data points illustrate microbial alpha diversity, assessed using the Shannon index (**left**) and the Inverse Simpson index (**right**). Individual data points represent the alpha diversity values of each participant, providing a visualization of within-group variability.

**Figure 4 nutrients-17-03909-f004:**
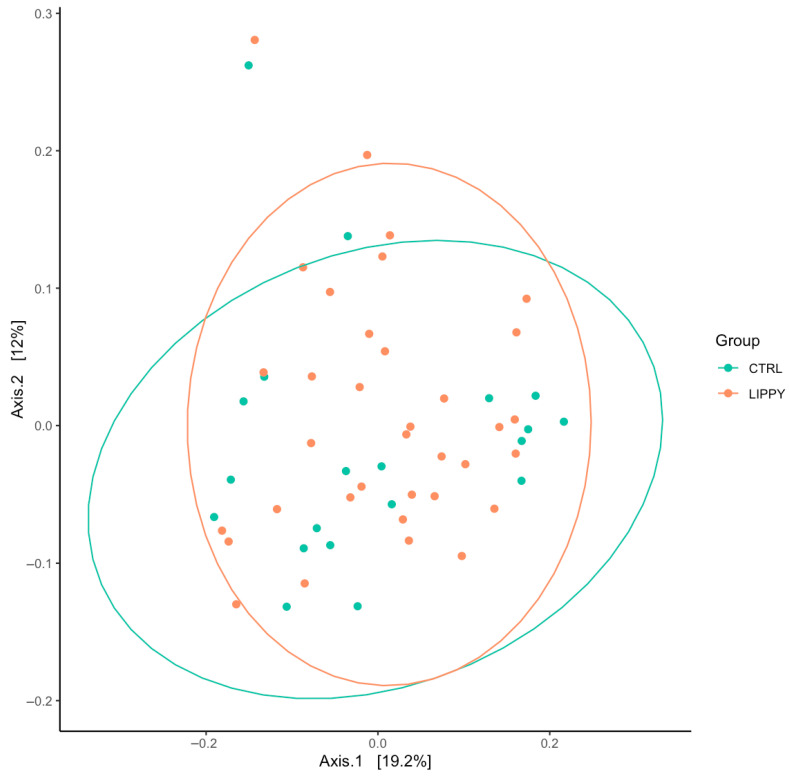
Beta diversity analysis using PcoA plots based on Bray–Curtis and Jaccard dissimilarity.

**Figure 5 nutrients-17-03909-f005:**
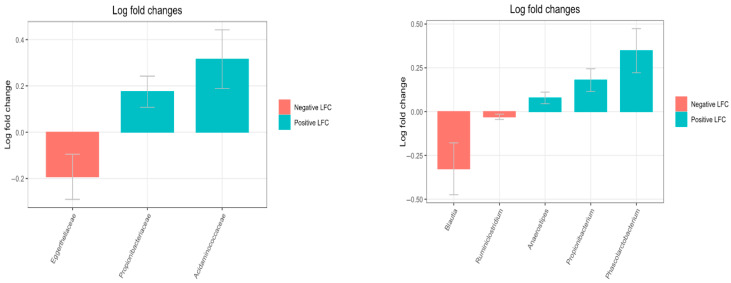
Differentially abundant taxa between LIPPY and CTRL groups. Bar plots display log_2_ fold changes for families (**left**) and genera (**right**). Taxa with *p* < 0.05 (ANCOM-BC) are included.

**Figure 6 nutrients-17-03909-f006:**
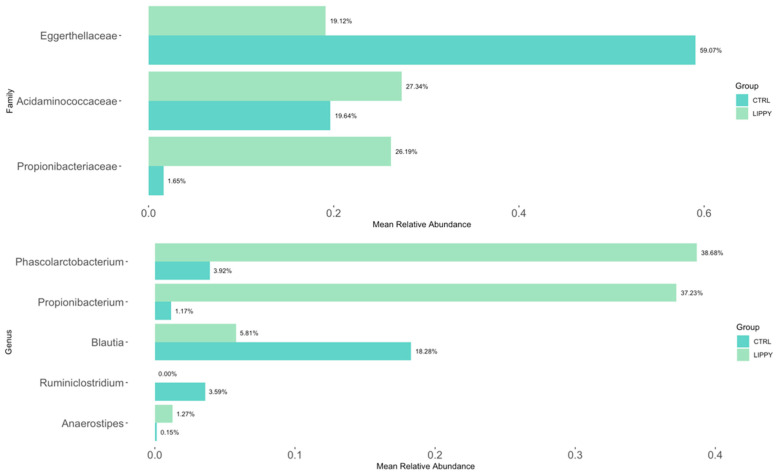
Differentially abundant taxa between LIPPY and CTRL groups. Bar plots display mean relative abundance at family and genus levels.

**Figure 7 nutrients-17-03909-f007:**
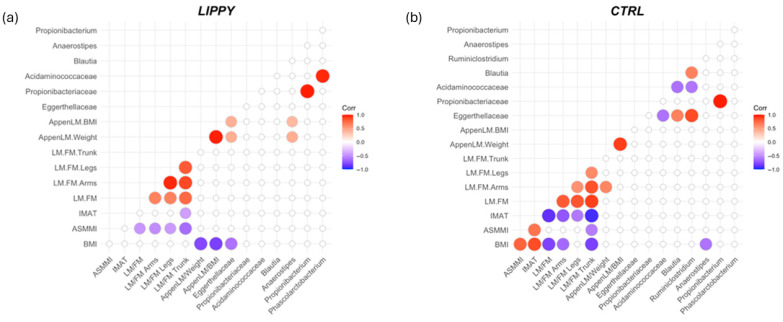
Heatmaps of Spearman correlations between microbial taxa (differentially abundant) and body composition parameters across (**a**) LIPPY group, and (**b**) CTRL group. Only correlations with *p* < 0.05 are shown.

**Figure 8 nutrients-17-03909-f008:**
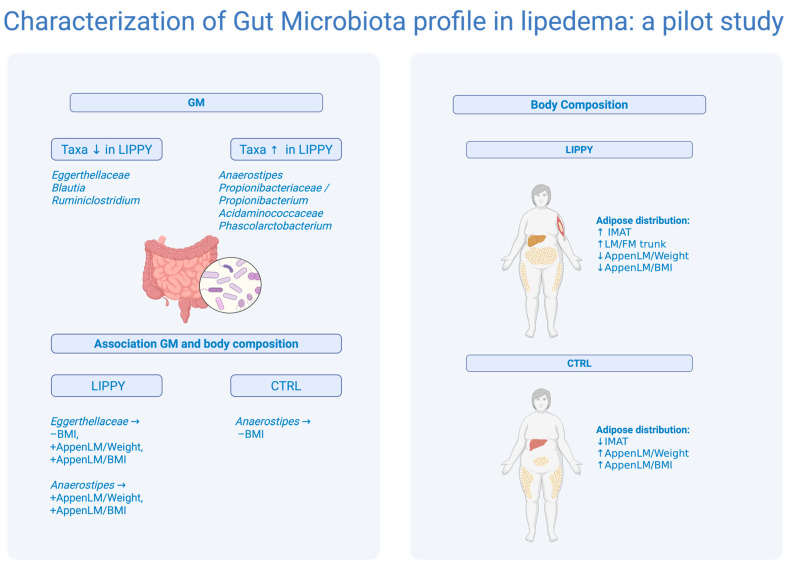
Summary of main results. Differences in body composition are highlighted, showing an increase in IMAT and a reduction in lean mass indices compared to fat mass in the LIPPY group. In addition, the main microbial taxa that differ between groups and their associations with BMI and appendicular lean mass indices are reported. Legend: ↑, increased; ↓, decreased; →, correlated.

**Table 1 nutrients-17-03909-t001:** Anthropometric and body composition characteristics of the study population.

	CTRL (Mean ± SD)	LIPPY (Mean ± SD)	Δ%	*p*-Value
BMI (kg/m^−2^)	21.20 ± 2.8	34.50 ± 10.7	+63%	7.38 × 10^−6^
ASMMI (kg/m^−2^)	6.19 ± 0.80	7.77 ± 1.56	+26%	5.06 × 10^−4^
IMAT (kg)	0.63 ± 0.25	1.07 ± 0.46	+70%	3.78 × 10^−3^
LM/FM trunk	2.83 ± 1.21	1.58 ± 0.94	−44%	1.72 × 10^−3^
LM/FM	2.32 ± 0.75	1.42 ± 0.87	−39%	6.16 × 10^−4^
LM/FMArms	1.91 ± 0.91	1.29 ± 0.52	−32%	4.47 × 10^−3^
LM/FMLegs	1.87 ± 0.51	1.08 ± 0.39	−42%	1.95 × 10^−5^
AppenLM/Weight	0.29 ± 0.03	0.24 ± 0.07	−17%	5.27 × 10^−3^
AppenLM/BMI	0.77 ± 0.13	0.62 ± 0.18	−19%	9.19 × 10^−3^

Values are presented as mean ± standard deviation (SD) for the Lipoedema (LIPPY) and control (CTRL) groups. The percentage change (Δ%) indicates the relative difference between groups, calculated as the percentage increase or decrease from the CTRL to the LIPPY group. Statistical significance was assessed using the two-tailed Wilcoxon rank-sum (Mann-Whitney U) test. Significant differences (*p* < 0.05) are highlighted. BMI: body mass index; ASMMI: appendicular skeletal muscle mass index; IMAT: intramuscular adipose tissue; LM/FM: lean mass to fat mass ratio; AppenLM: appendicular lean mass.

## Data Availability

The data presented in this study are available on request from the corresponding author. The data are not publicly available due to privacy.

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
