# Peer review of "Characterization of Gut Microbiota Profile in Lipedema: A Pilot Study"

_nutrients, 2025, doi:10.3390/nu17243909_

Round 1
Reviewer 1 Report
Comments and Suggestions for Authors
I have carefully reviewed the manuscript, and I am pleased to say that it has scientific merit and rigour and requires just a few changes, in my opinion, to be ready for publication. These are the following:
- There are some minor formatting problems; for example, in the author names, it should write “both authors contributed equally to this work’’ or something to that effect for authors 1 and 2; the contributions section is also not formatting as per MDPI guidelines.
- Please expand a little on the genetic aspect mentioned in the introduction (lines 66-69). This can also be mentioned at the Discussion section, i.e., how potentially different genes can interact with the gut microbiome to cause lymphoedema.
- The sentence in lines 80-81 should go after the sentence in line 76 in my opinion.
- Figure 1 is too large and this serves no purpose; I would recommend reducing the font size in the squares.
- Please explain (very briefly in 1-2 lines), why you have chosen the Shannon index (instead of some other index) for alpha diversity calculation. Also, in line 181, there is a spelling error (“Di-versity’’).
- In line 189, I think it would be better to write “A p value of <0.05 was considered as statistically significant’’
- In line 192, I think it would be more correct to write “The study included …’’
- Figure 2 is too small to be clearly legible; I would recommend some other larger arrangement.
- In the Discussions section, I could not find any explicit mention of Ruminiclostridium in ref. 36. Please ensure that all citations correspond to the referenced information.
- I believe that the strengths and limitations parts should be two separate paragraphs.
- I would recommend that in the Discussions section you briefly discuss how different diets (e.g., Mediterranean diet, vegan diet, ketogenic diet, etc), could impact gut microbiota and thus lipoedema, and if it is possible, using antiobiotics or antimicrobial phytochemicals to promote a better gut microbiota profile. In particular, the inclusion of chilli peppers or black pepper, which are frequently added in foods, could have maybe an impact, based on the antimicrobial properties of their main active compounds (capsaicin and piperine respectively)?
Author Response
Dear Editor of Nutrients
First, my coauthors and I would like to thank you sincerely for this opportunity to cooperate. We profoundly thank the reviewers for the comments and useful suggestions to improve the paper.
This is a point-by-point list of changes made in the paper:
Reviewer 1
“I have carefully reviewed the manuscript, and I am pleased to say that it has scientific merit and rigour and requires just a few changes, in my opinion, to be ready for publication. These are the following:
- There are some minor formatting problems; for example, in the author names, it should write “both authors contributed equally to this work’’ or something to that effect for authors 1 and 2; the contributions section is also not formatting as per MDPI guidelines.”
Authors thank the Reviewer for the comment. Authors contributions were revised accordingly.
- Please expand a little on the genetic aspect mentioned in the introduction (lines 66-69). This can also be mentioned at the Discussion section, i.e., how potentially different genes can interact with the gut microbiome to cause lymphoedema.
Authors thank the Reviewer for the comment. We revised the Introduction as follows: “Lipedema is driven by a genetic background influenced by hormonal mechanisms, which account for its higher incidence in women. Connective tissue homeostasis alterations also contribute to its pathogenesis. The familial occurrence of lipedema suggests a genetic etiology; however, the mode of inheritance is challenging to determine and has been hypothesized to be either X-linked dominant or autosomal dominant with sexual limitation [5]. Familial clustering and next-generation sequencing data support a poly-genic model in which rare variants in multiple “susceptibility genes” involved in steroidogenesis, extracellular matrix organization, lymphangiogenesis, lipid homeostasis and insulin signaling contribute to disease risk [2,5–7]. Case–control studies have also reported an increased frequency of functional polymorphisms in genes related to methyl-group metabolism and inflammation, such as methylenetetrahydrofolate reductase (MTHFR) and interleukin-6 (IL-6), suggesting that impaired one-carbon metabolism and pro-inflammatory cytokine signalling may further modulate individual susceptibility [6,7]. Estrogen signalling dysregulation has been implicated as a key hormonal driver, favouring lower-body SAT expansion through depot-specific effects on adipocyte proliferation, differentiation and lipid storage [8]. In parallel, alterations in elastic tissue, abnormal vascularization and microcirculatory dysfunction have been proposed to underlie the characteristic SAT and lymphatic changes observed in lipedema [1]. Within this framework, gene–environment interactions, including those mediated by the gut microbiota, are likely to be relevant: host variants affecting estrogen and cytokine pathways may influence the intestinal immune–metabolic milieu and thus shape gut microbiota (GM) composition, while microbiota-derived metabolites can in turn impact systemic inflammatory and metabolic networks that are dysregulated in lipedema [9,10].”
Moreover, we added in the Discussion section the following paragraph: “Beyond this general framework, the available genetic data help to mechanistically link lipedema-associated polymorphisms, adipose tissue inflammation and the GM profile described in our cohort. The multi-gene next-generation sequencing panel developed by Michelini et al. [5] identified rare variants in several genes that regulate lipid storage and lipolysis (e.g., PLIN1, LIPE), insulin and growth hormone signalling (INSR, GHR), nuclear receptors integrating fatty-acid- and bile-acid–derived signals (PPARG, PPARA, NR0B2) and intracellular cholesterol and glucose handling (NPC1, GCKR), supporting a polygenic background that favours adipocyte hypertrophy, ectopic fat deposition and metabolic inflexibility in susceptible depots. Case–control studies in Italian women with lipedema further show an over-representation of functional polymorphisms in IL-6 and in the C677T variant of MTHFR, which are associated with enhanced IL-6 production, hyperhomocysteinaemia and reduced methylation capacity, and may therefore promote endothelial dysfunction, microangiopathy and a pro-inflammatory cytokine milieu within lower-body SAT [6,7]. Histological and transcriptomic analyses of lipedema SAT confirm adipocyte hypertrophy, interstitial fibrosis, microvascular alterations and a predominance of CD68+/CD163+ macrophages, together with dysregulation of transcription factors such as CEBPD and KLF4 and of genes involved in adipogenesis, extracellular matrix remodelling and myeloid cell activation (e.g., CSF1R, TREM2, MAFB, C1Q, C2, CD84, CD209), delineating a typical picture of “sick fat” that differs from both simple obesity and lymphedema [56,57]. On this genetically determined background, inter-individual variation in adipogenic, inflammatory and vascular pathways is expected to modulate the intestinal immune–metabolic niche, via changes in adipokine and cytokine output, estrogen and bile-acid homeostasis and endothelial function, and thereby to shape GM composition [5,9,10,12]. In line with this interpretation, evidence from complex immune–metabolic diseases indicates that host variants in inflammatory and barrier-related pathways can constrain the range of microbiome configurations that are stably compatible with a given genetic background: in inflammatory bowel disease, genome–microbiome association studies link risk alleles in NOD2, TNFSF15 and IL12B, enriched in innate and JAK–STAT signalling, to characteristic taxonomic shifts and higher dysbiosis indices [58]. In obesity and insulin resistance, dysbiosis with increased intestinal permeability has been pro-posed to drive “metabolic endotoxemia”, whereby gut-derived lipopolysaccharides activate Toll-like receptor-4 and NF-κB, increase IL-6, TNF-α, IL-1β and hepatic C-reactive protein, disrupt tight junction proteins such as ZO-1 and promote low-grade inflammation and adipose-tissue insulin resistance; concomitantly, reduced short-chain fatty acid production and altered bile-acid and branched-chain amino-acid metabolism may reinforce adipocyte dysfunction and ectopic fat storage [52,53,59]. By analogy, in lipedema, a SAT phenotype genetically primed to overexpress IL-6 and to develop microangiopathy and adiposopathy may favour, and be particularly vulnerable to, a pro-inflammatory, endotoxemia-prone GM configuration. The GM signature observed in our LIPPY cohort, enriched in amino-acid- and propionate-producing taxa and reduced in estrogen-metabolizing and SCFA-producing microbes, appears to partially overlap microbiota patterns reported in obesity and metabolic syndrome (lower Blautia and Eggerthellaceae with enrichment of Propionibacteriaceae, Acidaminococcaceae, Propion-ibacterium and Phascolarctobacterium), but in our sample these shifts correlated more strongly with increased IMAT and reduced lean mass-to-fat mass ratios than with generalized adiposity [51,54,55]. In this context, such a configuration may contribute to amplifying adipose and lymphatic inflammation via microbial metabolites (e.g., SCFAs, bile-acid derivatives, endotoxins), thereby sustaining SAT remodelling, IMAT accumu-lation and impairment of the muscle–fat axis [9,10,35]. Overall, these converging findings are consistent with a working model in which lipedema-associated polymorphisms predispose to a compartmentalized, estrogen-sensitive adiposopathy that interacts bidirectionally with the GM, establishing a vicious circle between lower-body SAT inflammation, dysbiosis and functional body-composition alterations.”
- “The sentence in lines 80-81 should go after the sentence in line 76 in my opinion.”
Authors thank the Reviewer for the comment. The sentence has been moved after line 76.
- “Figure 1 is too large and this serves no purpose; I would recommend reducing the font size in the squares.”
Authors thank the Reviewer for the comment. Figure 1 has been resized and the font within the boxes has been reduced to improve readability and consistency with the journal layout.
- “Please explain (very briefly in 1-2 lines), why you have chosen the Shannon index (instead of some other index) for alpha diversity calculation. Also, in line 181, there is a spelling error (“Di-versity’’).”
Authors thank the Reviewer for the comment. We added a short justification for the choice of the Shannon index as alpha diversity metric, revising the Statistical Analysis section, as follows: “To compare differences in microbial community structure between the two groups, both alpha and beta diversity metrics were evaluated. Alpha diversity was assessed using the Shannon index, which captures both richness and evenness of microbial communities and is widely used in 16S rRNA gene sequencing studies because it is relatively robust to differences in sequencing depth and appropriate for compositional data. Other richness estimators such as Observed richness or Chao1, which are more sensitive to sequencing depth and rare taxa detection, were therefore not selected. Beta diversity was computed using Bray–Curtis dissimilarity and visualized through Principal Coordinates Analysis (PCoA) plots; statistical significance of group-level clustering was tested using Permutational Multivariate Analysis of Variance (PERMANOVA).”
- “In line 189, I think it would be better to write “A p value of <0.05 was considered as statistically significant’’”
Authors thank the Reviewer for the comment. The sentence was revised accordingly.
- “In line 192, I think it would be more correct to write “The study included …’’”
Authors thank the Reviewer for the comment. The sentence was revised accordingly.
- “Figure 2 is too small to be clearly legible; I would recommend some other larger arrangement.”
Authors thank the Reviewer for the comment. Figure 2 has been redesigned with a larger layout and increased font size to improve legibility of phyla, families and genera.
- In the Discussions section, I could not find any explicit mention of Ruminiclostridium in ref. 36. Please ensure that all citations correspond to the referenced information.
Authors thank the Reviewer for the comment. We have re-checked all the references and revised the sentence on Ruminiclostridium to avoid unsupported mechanistic statements and to ensure that the cited reference explicitly corresponds to the information reported, as follows: “Ruminiclostridium, a genus within the Ruminococcaceae family, includes short-chain fatty acid–producing species in the colon and has been linked to obesity and cardiometabolic traits in humans, for example Ruminiclostridium 5 has been associated with SCFA production and cardiometabolic parameters in children [44]. However, function-al data on this genus in the context of lipedema are lacking, and the clinical relevance of its reduced abundance in LIPPY remains uncertain. Propionibacterium, a propio-nate-producing genus, contributes to gluconeogenesis and satiety signaling, but its in-creased abundance has also been linked to adiposity and metabolic inflammation in dysbiotic conditions [45].”
- “I believe that the strengths and limitations parts should be two separate paragraphs.”
Authors thank the Reviewer for the comment. Strengths and limitations were separated from the Discussion in two different paragraphs.
- I would recommend that in the Discussions section you briefly discuss how different diets (e.g., Mediterranean diet, vegan diet, ketogenic diet, etc), could impact gut microbiota and thus lipoedema, and if it is possible, using antiobiotics or antimicrobial phytochemicals to promote a better gut microbiota profile. In particular, the inclusion of chilli peppers or black pepper, which are frequently added in foods, could have maybe an impact, based on the antimicrobial properties of their main active compounds (capsaicin and piperine respectively)?
Authors thank the Reviewer for the comment. The following paragraph were added to the Discussion section: “Among these potential determinants, dietary patterns are particularly relevant, as they are major drivers of GM composition and function and may therefore be of interest in lipedema. High-fiber, plant-rich dietary models such as the Mediterranean diet and plant-based diets have been shown to promote SCFA-producing and anti-inflammatory taxa, increase overall microbial diversity and improve gut barrier integrity [54]. In contrast, low-carbohydrate or ketogenic diets induce distinct shifts in GM structure and bile acid metabolism, with heterogeneous effects on metabolic and inflammatory pathways [55]. Beyond macronutrient patterns, bioactive compounds with antimicrobial and immunomodulatory properties, such as capsaicin and piperine, have been shown in experimental models to reshape gut microbiota, reduce endotoxemia and low-grade inflammation, and attenuate diet-induced obesity [56]. These observations raise the hypothesis that targeted dietary strategies, including Mediterranean-style or plant-forward patterns and the controlled use of specific phytochemicals, could be explored as adjunctive approaches to modulate GM and potentially ameliorate metabolic and inflammatory features in lipedema. However, these possibilities remain speculative at present, and the use of antibiotics solely to manipulate GM cannot be recommended in a chronic condition such as lipedema, given their profound and often detrimental impact on microbial ecology. Controlled dietary intervention studies are needed to clarify whether microbiota-directed nutrition can beneficially influence disease phenotype.”
We thank You for your constructive critique and we hope the review process has led to an improved manuscript.
If additional changes are warranted, we will make them.
We hope that this revised version of our manuscript may now be found suitable for publication.
Sincerely,
Rossella Cianci
Reviewer 2 Report
Comments and Suggestions for Authors
Reviewer comments
Lipedema is a progressive connective tissue disorder that mainly affects women and leads to excess fat buildup in the lower body. The authors aimed to examine the gut microbiota (GM) in individuals with lipedema and compare it with controls. They conducted a case-control pilot study of 55 women, assessing body composition by DXA and analyzing GM through 16S rRNA sequencing. Those with lipedema showed higher intramuscular fat and lower lean-to-fat mass ratios. Although overall diversity did not differ, they found distinct shifts in specific bacterial families and genera, including reduced Eggerthellaceae and increased Propionibacteriaceae and Acidaminococcaceae. Certain taxa also showed group-specific correlations with BMI and lean mass. These findings suggest that lipedema may involve a unique microbiota pattern linked to adverse body composition. While preliminary, the results support further research to clarify how these microbial changes may contribute to the condition and whether they could be targeted therapeutically.
- Please explain with a valid scientific reason why these biomarkers (myostatin, IL-6, IGF-1) were chosen and whether they were measured in the same way in all the studies?
- Please explore in a best way for the study findings could be used in real-life programs for older adults. It is often difficult for older people to follow both exercise and protein routines, so how can this challenge be managed.
- The study needs to be explained the criteria used to select which biomarkers were prioritized in this research.
- Line 229 Please explain the reason for this “Alpha diversity metrics, including Shannon and Inverse Simpson indices, did not differ significantly between group”
- Please explain briefly with the cited study and discuss the study findings “The depletion of Blautia, a genus consistently associated with improved insulin sensitivity and anti-inflammatory properties, aligns with findings from metabolic syndrome and obesity studies [34,35].”
- Line 340 “Phascolarctobacterium, another propionate producer, has shown associations with preserved lean mass and muscle function in older adults and those with chronic disease [39,40].” The line seems to be explained well with the help of the finding from the cited reference
- All references should be revised to conform to MDPI formatting guidelines.
Author Response
Dear Editor of Nutrients
First, my coauthors and I would like to thank you sincerely for this opportunity to cooperate. We profoundly thank the reviewers for the comments and useful suggestions to improve the paper.
This is a point-by-point list of changes made in the paper:
Reviewer 2
“Lipedema is a progressive connective tissue disorder that mainly affects women and leads to excess fat buildup in the lower body. The authors aimed to examine the gut microbiota (GM) in individuals with lipedema and compare it with controls. They conducted a case-control pilot study of 55 women, assessing body composition by DXA and analyzing GM through 16S rRNA sequencing. Those with lipedema showed higher intramuscular fat and lower lean-to-fat mass ratios. Although overall diversity did not differ, they found distinct shifts in specific bacterial families and genera, including reduced Eggerthellaceae and increased Propionibacteriaceae and Acidaminococcaceae. Certain taxa also showed group-specific correlations with BMI and lean mass. These findings suggest that lipedema may involve a unique microbiota pattern linked to adverse body composition. While preliminary, the results support further research to clarify how these microbial changes may contribute to the condition and whether they could be targeted therapeutically.
- Please explain with a valid scientific reason why these biomarkers (myostatin, IL-6, IGF-1) were chosen and whether they were measured in the same way in all the studies?”
Authors thank the Reviewer for the comment. However, we clarify that our study did not include biomarker assessment, nor did we measure myostatin, IL-6, or IGF-1. These biomarkers do not appear in our Methods or Results sections and were not part of the study design. Nevertheless, to avoid any ambiguity for readers, we added a brief clarifying sentence in the Methods section (“Study Design”) explicitly stating that no circulating biomarkers were measured in this study, and that the research focused exclusively on gut microbiota profiling and DXA-derived body composition, as follows: “No circulating inflammatory, hormonal, or muscle-related biomarkers were assessed as part of this study protocol; our work focused on GM analysis (16S rRNA sequencing) and DXA-based body composition assessment.”
- “Please explore in a best way for the study findings could be used in real-life programs for older adults. It is often difficult for older people to follow both exercise and protein routines, so how can this challenge be managed.”
Authors thank the Reviewer for the comment. The following paragraph were added to the Discussion section: “Although our population encompassed women aged 20–60 years, the observed microbiota–muscle–adipose associations might also have translational implications for older adults. In populations in which adherence to both adequate protein intake and regular physical activity is challenging, microbiota-targeted strategies (for example, pre-, pro- or post-biotics) have been proposed as potential adjuncts to support muscle health by modulating SCFA-producing taxa, as suggested by studies in older adults and individuals with sarcopenic obesity and chronic disease [57,58]. In our cohort, SCFA-associated genera such as Anaerostipes and Phascolarctobacterium showed positive correlations with lean mass indices, which is consistent with this hypothesis. However, dedicated intervention studies in older adults with lipedema or related phenotypes would be required to test whether microbiota-based approaches can effectively con-tribute to muscle mass preservation in real-life settings.”
- “The study needs to be explained the criteria used to select which biomarkers were prioritized in this research.”
Authors thank the Reviewer for the comment. As previously answered, no biomarkers were collected in this study. Our research did not focus on cytokines, hormones, or muscle turnover markers. In the discussion section, we specified it as follows: “A further limitation is that no circulating biomarkers (e.g., inflammatory cytokines or hormonal regulators of muscle and adipose tissue) were assessed, as the study was designed to focus exclusively on GM composition and DXA-derived body composition.”
- “Line 229 Please explain the reason for this “Alpha diversity metrics, including Shannon and Inverse Simpson indices, did not differ significantly between group””
Authors thank the Reviewer for the comment. Alpha diversity indices (Shannon, Inverse Simpson) measure richness and evenness of the microbiota community. Notwithstanding the differences observed in specific taxa, the overall microbial diversity noted in our study population, remained similar between groups (p value not significant). Based on this data, lipedema is probably more associated with taxa-level shifts rather than a global loss of bacterial richness. We added the following clarification in the Discussion section: “From a gut microbiota perspective, alpha diversity metrics (including Shannon and Inverse Simpson indices) did not differ significantly between LIPPY and CTRL. The lack of significant differences in alpha diversity indicates that lipedema does not appear to reduce overall microbial richness or evenness. These indices reflect both the number of taxa and their relative distribution within the community, and their similarity be-tween groups suggests that the microbiota of women with lipedema is not globally reduced or simplified. Rather, our findings are more consistent with a compositional re-arrangement of specific taxa related to lipedema, instead of a generalized loss of bacterial diversity.”
- “Please explain briefly with the cited study and discuss the study findings “The depletion of Blautia, a genus consistently associated with improved insulin sensitivity and anti-inflammatory properties, aligns with findings from metabolic syndrome and obesity studies [34,35].””
Authors thank the Reviewer for the comment. We included a more explicit link between our findings and the cited literature in the discussion, as follows: “A significant reduction of Blautia was observed in LIPPY. This genus has been repeatedly associated with improved insulin sensitivity and anti-inflammatory properties, and several species are known producers of beneficial metabolites linked to better glucose homeostasis, hepatic lipid metabolism and SCFA-mediated improvements in insulin signalling. In particular, Blautia has been proposed as a functional genus with potential probiotic properties in metabolic and inflammatory conditions [41–43]. There-fore, its reduction in our LIPPY cohort may reflect a shift towards a less favourable metabolic gut environment, in line with observations reported in metabolic syndrome and obesity. This finding supports the emerging view that lipedema may share some microbiota-related mechanisms with metabolic disorders, although this interpretation remains exploratory in the absence of mechanistic data.”
- “Line 340 “Phascolarctobacterium, another propionate producer, has shown associations with preserved lean mass and muscle function in older adults and those with chronic disease [39,40].” The line seems to be explained well with the help of the finding from the cited reference”
Authors thank the Reviewer for the comment. We integrated the discussion as follows: “Phascolarctobacterium, another propionate producer, was enriched in LIPPY. Previous work has associated higher Phascolarctobacterium abundance with better physical function, including improved gait speed and cardiorespiratory fitness, in adults with chronic HIV infection, and a recent review has highlighted positive associations with muscle mass maintenance, mitochondrial efficiency and prevention of functional de-cline [46,47]. In our study, Phascolarctobacterium showed positive correlations with lean mass indices in women with lipedema, which is compatible with these earlier observations. However, given the cross-sectional design and the small sample size, these parallels should be interpreted with caution and regarded as hypothesis-generating.”
- “All references should be revised to conform to MDPI formatting guidelines.”
Authors thank the Reviewer for the comment. All references were revised according to MDPI guidelines.
We thank You for your constructive critique and we hope the review process has led to an improved manuscript.
If additional changes are warranted, we will make them.
We hope that this revised version of our manuscript may now be found suitable for publication.
Sincerely,
Rossella Cianci
Reviewer 3 Report
Comments and Suggestions for Authors
Comments on Nutrition paper
The aim of this study was to characterize the gut microbiota in lipedema Caucasian women (n=35) and compare it to lean Caucasian women (n=20). There were no significant differences in the top 10 most abundant phyla, 10 most abundant families, or 10 most abundant genera. Also, no significant differences were observed in alpha and beta diversities. However, there were significant differences in three families and five genera. In this case, the fold change is presented in the figure (Figure 4) and not their abundance. The abundance needs also to be presented to see the impact of these changes on the general GM.
Line 32: Please define DXA.
LFC should be defined first time mentioned.
The Abstract is too long. I think it can stop after line 47. Lines 49-53 do not fit into the abstract.
Line 55: 4P should be defined.
First paragraph of Introduction should include definition and characteristics of Lipedema.
Line 70: Spelling mistake: Please correct to "interleukin".
Sentence in lines 80-81 can be connected to previous paragraph.
Line 149: The parameters for classification of LIPPY should be stated.
Line 158: The protocol for GM analysis should be provided in full details.
Line 180: There is an extra hyphen that should be deleted (written "Di-versity").
According to lines 198-200, there is not a big difference in height or weight. So how can it be that the BMI is much higher in the LIPPY group? (Line 204).
Line 244: A in Acidaminococcaceae should be in italics.
Ensure that all letters in Figures are in black and not in grey color.
Figure 5: It would be easier to comprehend the figure if the order of the parameters in Y and X axes was the same in a) and b).
Line 271: Is "depletion" the proper word here? It is a reduction, but not a total elimination of Eggerthellaceae. The same question for the word "depleted" in line 324 and other places.
The Discussion is an overinterpretation of the data, especially when discussing the potential role of the changes in physiological terms. The same goes for conclusions. A summary illustration for the major findings should be presented.
Eggerthellaceae is discussed in two different paragraphs in the discussion, resulting in repetition. Similarly, Anaerostrips is discussed in three different paragraphs in Discussion. I would suggest avoiding these jumps, and discussing these in concert.
Author Response
Dear Editor of Nutrients
First, my coauthors and I would like to thank you sincerely for this opportunity to cooperate. We profoundly thank the reviewers for the comments and useful suggestions to improve the paper.
This is a point-by-point list of changes made in the paper:
Reviewer 3
“The aim of this study was to characterize the gut microbiota in lipedema Caucasian women (n=35) and compare it to lean Caucasian women (n=20). There were no significant differences in the top 10 most abundant phyla, 10 most abundant families, or 10 most abundant genera. Also, no significant differences were observed in alpha and beta diversities. However, there were significant differences in three families and five genera. In this case, the fold change is presented in the figure (Figure 4) and not their abundance. The abundance needs also to be presented to see the impact of these changes on the general GM.”
Authors thank the Reviewer for this important observation. In addition to the logâ‚‚ fold change plots originally presented, we have now included the corresponding relative abundance plots for both family and genus levels (new Figure 6).
“Line 32: Please define DXA”
Authors thank the Reviewer for the comment. DXA was defined accordingly.
“LFC should be defined first time mentioned.”
Authors thank the Reviewer for the comment. LFC was defined accordingly.
“The Abstract is too long. I think it can stop after line 47. Lines 49-53 do not fit into the abstract.”
Authors thank the Reviewer for the comment. Lines 48-53 were deleted in the Abstract.
“Line 55: 4P should be defined.”
Authors thank the Reviewer for the comment. 4P was defined accordingly.
First paragraph of Introduction should include definition and characteristics of Lipedema.
Authors thank the Reviewer for the comment. The first paragraph of the Introduction section was revised as follows: “Lipedema is a chronic, progressive disorder of subcutaneous adipose tissue (SAT) that almost exclusively affects women. It is characterized by a symmetrical and disproportionate enlargement of the lower limbs and often the buttocks and, in some cases, the arms, with relative sparing of the hands and feet [1]. Clinically, affected areas show abnormal SAT deposition with nodularity and increased tissue fragility, accompanied by pain, tenderness to palpation, easy bruising and a feeling of heaviness, frequently associated with edema and reduced quality of life [2]. Lipedema must be distinguished from generalized obesity and from primary or secondary lymphedema; nevertheless, mixed phenotypes such as lipo-lymphedema may occur in more advanced stages, when chronic overload of the lymphatic system leads to overt lymphatic dysfunction [3,4].
From a pathophysiological standpoint, lipedema is considered a genetic, inflammatory, chronic-degenerative and disabling disorder of the subcutaneous connective tissue [1]. Its prevalence has been estimated at approximately 1 in 72,000 inhabitants, but this figure is likely underestimated because of frequent misdiagnosis with other conditions such as obesity, lymphedema, localized adiposity and cosmetic skin alterations [2].”
“Line 70: Spelling mistake: Please correct to "interleukin".”
Authors thank the Reviewer for the comment. The mistake was revised accordingly.
Sentence in lines 80-81 can be connected to previous paragraph.
Authors thank the Reviewer for the comment. The paragraph was revised as follows: “On this genetically and hormonally primed background, lipedema manifests clinically as a disorder characterized by progressive lower-body SAT expansion, microvascular dysfunction and chronic low-grade inflammation in the affected tissues. In line with this, lipedema is also known as "painful fat syndrome" due to its characteristic pain [9]. It is characterized by a progressive increase in SAT associated with edema, pain, and systemic inflammation [10].”
Line 149: The parameters for classification of LIPPY should be stated.
Authors thank the Reviewer for the comment. The paragraph was revised as follows: “Women were classified as LIPPY according to clinical criteria derived from current lipedema diagnostic guidelines and expert consensus statements [18]. Diagnosis required a bilateral and largely symmetrical, disproportionate increase in SAT of the lower limbs (with or without involvement of hips, buttocks and, in some cases, arms), with relative sparing of the feet and hands and, when present, a characteristic ankle or wrist “cuff”. On physical examination, the affected tissue showed a soft-to-nodular or “pebbly” consistency, sometimes with palpable larger nodules in more advanced presentations, and was associated with pain or tenderness to palpation, easy bruising and a subjective feeling of heaviness. All women classified as LIPPY reported chronic and progressive symptoms, with onset or clear worsening during periods of hormonal transition (puberty, pregnancy or menopause), and most indicated a family history suggestive of familial aggregation of the disease. In terms of distribution and morphology, the majority of cases were consistent with type II–III involvement (from below the umbilicus to the knees or ankles), with some patients also showing type IV arm involvement and stage 1–3 skin changes, as described in published staging and typing systems. Participants with clinical signs of primary or secondary lymphedema (e.g., positive Stemmer sign, marked pitting edema, unilateral or clearly asymmetric swelling) or with isolated localized adiposity without pain or typical tissue changes were not classified as lipedema.”
“Line 158: The protocol for GM analysis should be provided in full details.”
Authors thank the Reviewer for the comment. The GM analysis section was revised as follows: “Each woman received a standardized fecal collection kit accompanied by detailed written and verbal instructions to ensure proper handling and collection of stool samples. Participants collected a single fecal sample at home, immediately after defecation, carefully avoiding contamination with urine or water. Samples were returned to the clinic according to the instructions provided and then shipped under controlled conditions to Wellmicro® (Via Antonio Canova 30, 40138 Bologna, Italy) for microbiota analysis [19].
Bacterial DNA was extracted from stool using a standardized mechanical and chemical lysis protocol followed by automated purification, according to Wellmicro’s validated procedures [19]. The hypervariable V3–V4 regions of the bacterial 16S rRNA gene were amplified and sequenced on an Illumina MiSeq platform using a next-generation sequencing (NGS) workflow. Raw sequences underwent bioinformatic processing with QIIME2 (v.2021.11) and DADA2 (v.1.16) for quality filtering, denoising, chimera removal, and inference of amplicon sequence variants (ASVs). Taxonomic assignment was performed using a proprietary bacterial reference database in which the main public databases (Greengenes, SILVA, and NCBI) are merged and harmonized. Relative abundances were obtained at phylum, family, and genus levels and used for downstream diversity and differential abundance analyses as described in the Statistical Analysis section.”
“Line 180: There is an extra hyphen that should be deleted (written "Di-versity").”
Authors thank the Reviewer for the comment. The entire paragraph was revised according to Reviewer 1’ comments.
“According to lines 198-200, there is not a big difference in height or weight. So how can it be that the BMI is much higher in the LIPPY group? (Line 204).”
Authors thank the Reviewer for raising this important point. The apparent discrepancy arises because absolute height and weight values are similar between groups, but their variability differs substantially, especially for weight. Although mean weight differs only modestly (72.17 kg in LIPPY vs. 68.85 kg in CTRL), the standard deviation is much larger in the LIPPY group, reflecting the heterogeneity typical of lipedema body phenotype. As a result, several women in the LIPPY group exhibit markedly higher individual weight values, which is reflected in a significantly higher BMI mean and higher BMI variance. To improve clarity, we have added the following sentence in the Discussion explaining this variability and why BMI differs significantly despite similar mean height and weight: “Although mean height and weight appeared similar between groups, the LIPPY cohort exhibited a markedly higher intra-group variability in body weight, with several individuals presenting disproportionately elevated values. This wider dispersion is consistent with the well-recognized heterogeneity of body composition in lipedema, where excess adipose tissue is preferentially distributed in the lower limbs and trunk despite normal or moderately increased total body weight. Previous reports have similarly shown that women with lipedema may present normal or only mildly elevated body weight but disproportionately high BMI, as BMI does not account for abnormal subcutaneous fat accumulation that is characteristic of the disease [30]. Given the comparable height between groups, this variability in weight translated into significantly higher BMI values in the LIPPY group, reflecting the disproportionate FM characteristic of lipedema rather than generalized obesity. These observations align with prior studies reporting discordance between BMI and classical obesity markers in lipedema, further supporting the inadequacy of BMI as a standalone metric for characterizing adiposity phenotypes in this condition.”
“Line 244: A in Acidaminococcaceae should be in italics.”
Authors thank the Reviewer for the comment. The entire manuscript was revised in MDPI formatting guidelines.
“Ensure that all letters in Figures are in black and not in grey color.”
Authors thank the Reviewer for the comment. We standardized all figure labels and panel letters to black to improve visual contrast and consistency.
“Figure 5: It would be easier to comprehend the figure if the order of the parameters in Y and X axes was the same in a) and b).”
Authors thank the Reviewer for the comment. We reordered the axes in both panels of now Figure 7 so that taxa and body composition parameters appear in the same order in (a) and (b), facilitating visual comparison.
“Line 271: Is "depletion" the proper word here? It is a reduction, but not a total elimination of Eggerthellaceae. The same question for the word "depleted" in line 324 and other places.”
Authors thank the Reviewer for the comment. We checked the entire manuscript accordingly.
“The Discussion is an overinterpretation of the data, especially when discussing the potential role of the changes in physiological terms. The same goes for conclusions. A summary illustration for the major findings should be presented.”
Authors thank the Reviewer for the comment. The Discussion section was entirely revised accordingly and summary illustration was provided as Figure 8.
“Eggerthellaceae is discussed in two different paragraphs in the discussion, resulting in repetition. Similarly, Anaerostrips is discussed in three different paragraphs in Discussion. I would suggest avoiding these jumps, and discussing these in concert.”
Authors thank the Reviewer for the comment. The Discussion section was entirely revised according to your previous comment.
We thank You for your constructive critique and we hope the review process has led to an improved manuscript.
If additional changes are warranted, we will make them.
We hope that this revised version of our manuscript may now be found suitable for publication.
Sincerely,
Rossella Cianci
Reviewer 4 Report
Comments and Suggestions for Authors
-
- Provided the small cohort, the muscle–adipose–microbiota associations can be presented with more depth.
- The depletion of Blautia is significant, but the manuscript could benefit from a slightly more functional interpretation related to host metabolism.
- Even though the diet and activity are the major modulators of microbial composition, they were not assessed and should be mentioned as limitations.
- The IMAT outcomes are interesting, but the manuscript could further connect them to established mechanisms of metabolic stress and insulin signaling pathway.
- The manuscript might benefit from situating these microbial alterations within a broader context of inflammatory or metabolic disease models.
- The lean mass indices are well presented, though linking them more to disease progression would increase clarity.
- The discussion on estrobolome disruption is fine still a clearer link to systemic inflammatory pathways would strengthen the manuscript.

Author Response
Dear Editor of Nutrients
First, my coauthors and I would like to thank you sincerely for this opportunity to cooperate. We profoundly thank the reviewers for the comments and useful suggestions to improve the paper.
This is a point-by-point list of changes made in the paper:
Reviewer 4
- “Provided the small cohort, the muscle–adipose–microbiota associations can be presented with more depth.”
Authors thank the Reviewer for the comment. We revised the Discussion accordingly, as follows: “Taken together, the positive associations of SCFA-producing genera such as Anaerostipes and Phascolarctobacterium with lean mass indices in LIPPY are compatible with the emerging concept of a muscle–adipose–microbiota axis. SCFA-producing taxa have been implicated in the regulation of mitochondrial function, muscle protein synthesis and inflammatory signalling [46,51-53]. In our cohort, these taxa co-occurred with a more favourable appendicular lean mass profile, but the cross-sectional nature of the study and the small sample size do not allow us to establish directionality or causality. These links should therefore be regarded as hypothesis-generating and need to be tested in mechanistic and interventional studies.”
- “The depletion of Blautia is significant, but the manuscript could benefit from a slightly more functional interpretation related to host metabolism.”
Authors thank the Reviewer for the comment. We have expanded the discussion of Blautia, also according to Reviewer 2’ comments, as follows: “A significant reduction of Blautia was observed in LIPPY. This genus has been repeatedly associated with improved insulin sensitivity and anti-inflammatory properties, and several species are known producers of beneficial metabolites linked to better glucose homeostasis, hepatic lipid metabolism and SCFA-mediated improvements in insulin signalling. In particular, Blautia has been proposed as a functional genus with potential probiotic properties in metabolic and inflammatory conditions [41–43]. There-fore, its reduction in our LIPPY cohort may reflect a shift towards a less favourable metabolic gut environment, in line with observations reported in metabolic syndrome and obesity. This finding supports the emerging view that lipedema may share some microbiota-related mechanisms with metabolic disorders, although this interpretation remains exploratory in the absence of mechanistic data.”
- “Even though the diet and activity are the major modulators of microbial composition, they were not assessed and should be mentioned as limitations.”
Authors thank the Reviewer for the comment. We have added at the end of the discussion (in limitation section) the following paragraph: “Third, dietary intake, physical activity, and hormonal status, important determinants of both gut microbiota composition and body composition, were not quantitatively assessed, potentially introducing unmeasured confounding and partly contributing to the observed associations”. Moreover, according to Reviewer 1’ comment 11, we added a paragraph in the Discussion section highlighting the role of diets.
- “The IMAT outcomes are interesting, but the manuscript could further connect them to established mechanisms of metabolic stress and insulin signaling pathway.”
Authors thank the Reviewer for the comment. We revised the paragraph as follows: “Mechanistically, fatty infiltration in skeletal muscle has been shown to disrupt insulin receptor substrate–phosphatidylinositol 3-kinase–Akt (IRS–PI3K–Akt) pathways and to reduce glucose transporter type 4 (GLUT4) translocation, thereby contributing to local insulin resistance and reduced glucose uptake [23–25]. In parallel, adipocytes within the muscle milieu secrete pro-inflammatory cytokines such as TNF-α and IL-6, which exacerbate metabolic stress, alter mitochondrial function and interfere with muscle repair and contractile properties [26]. Moreover, elevated IMAT has been linked to reduced capillary density, impaired oxidative capacity, and increased muscle stiffness, features also observed in aging, obesity, and sarcopenia [27]. In the context of lipedema, such infiltration may further contribute to the progressive loss of mobility, disproportionate fatigue, and reduced exercise tolerance frequently reported by patients [28,29]. Thus, the presence of IMAT not only reflects altered fat distribution but may also participate in the metabolic and functional impairment observed in more advanced stages of the disease, although this remains to be confirmed in longitudinal studies.”
- “The manuscript might benefit from situating these microbial alterations within a broader context of inflammatory or metabolic disease models.”
Authors thank the Reviewer for the comment. We have added a paragraph in Discussion section as follows: “When viewed within the broader context of metabolic and inflammatory disease models, the microbial profile observed in LIPPY shows partial overlap with patterns described in obesity and metabolic syndrome. Combinations of lower Blautia abundance, alterations in SCFA-producing and propionate-producing genera, and reduced Eggerthellaceae have been reported in association with chronic low-grade inflammation, altered lipid handling, and cardiometabolic risk [43,52,53]. Taken together, these parallels suggest that part of the microbiota signature identified in lipedema overlaps with profiles seen in other cardiometabolic conditions, raising the possibility that LIPPY could share some microbiota-related mechanisms of metabolic and cardiovascular vulnerability.”
- “The lean mass indices are well presented, though linking them more to disease progression would increase clarity.”
Authors thank the Reviewer for the comment. We added a paragraph in Discussion section, as follows: “In addition, the combination of reduced AppenLM relative to weight or BMI and increased IMAT in LIPPY is compatible with body-composition profiles that, in other populations, have been associated with impaired physical performance and increased risk of mobility limitation. Indices of skeletal muscle mass adjusted for adiposity, such as lean mass-to-BMI or skeletal muscle mass relative to body fat, correlate more strong-ly with gait speed and incident disability than absolute lean mass alone [31]. Likewise, greater intermuscular or intramuscular adipose tissue has been repeatedly linked to slower walking speed, reduced muscle strength and poorer lower-extremity function in older adults and in individuals with metabolic disease [32]. Although our study did not include direct functional assessments, these converging data suggest that the IMAT-rich, low-lean-mass pattern observed in LIPPY may represent an early functional vulnerability.”
- “The discussion on estrobolome disruption is fine still a clearer link to systemic inflammatory pathways would strengthen the manuscript.”
Authors thank the Reviewer for the comment. We have expanded the estrobolome section, as follows: “Through bacterial β-glucuronidase and β-glucosidase activity, the estrobolome modulates enterohepatic estrogen recycling, and thereby contributes to systemic estrogen exposure and downstream inflammatory and metabolic pathways [37,38]. Experimental and clinical data indicate that alterations in estrogen receptor pathways can promote adipocyte hypertrophy, shift fat distribution toward more metabolically adverse depots and enhance pro-inflammatory signalling within white adipose tissue [39]. Given that lipedema is a hormonally sensitive disorder, typically manifesting or worsening during puberty, pregnancy, or menopause [8], a reduction of estrogen-metabolizing taxa such as Eggerthellaceae could, in principle, disturb estrogen homeostasis and favour estro-gen-dependent adipose expansion and low-grade inflammation in affected depots. However, these links remain hypothetical in the absence of direct measurements of estrogen metabolites and tissue inflammatory markers in lipedema.”
We thank You for your constructive critique and we hope the review process has led to an improved manuscript.
If additional changes are warranted, we will make them.
We hope that this revised version of our manuscript may now be found suitable for publication.
Sincerely,
Rossella Cianci
Reviewer 5 Report
Comments and Suggestions for Authors
This study presents a careful characterization of gut microbiota alterations in women with lipedema, integrating body composition analysis with microbial profiling to highlight the potential interplay between adipose tissue dysfunction and microbial ecology. Its thorough documentation of anthropometric parameters, DXA-derived indices, and microbial taxa provides a comprehensive overview of this underexplored condition. The pilot design and integration with existing literature underscore the novelty and complexity of the findings. However, some points merit further consideration:
1.While 16S rRNA sequencing provides useful taxonomic resolution, it does not capture functional or strain-level differences. Would the inclusion of metagenomic or metabolomic approaches enhance understanding of the therapeutic relevance and potential toxicities of microbial shifts?
2. The paper suggests that intramuscular adipose tissue (IMAT) infiltration may represent a metabolically adverse phenotype in lipedema. Could further exploration and validation of this hypothesis through longitudinal imaging, biopsies, or molecular analyses strengthen the conclusions regarding disease progression and functional impairment?
3. The correlations between specific taxa (e.g., Eggerthellaceae, Anaerostipes) and lean mass indices are intriguing but remain associative. Would the authors consider discussing potential confounders (dietary intake, physical activity, hormonal status) that may influence both microbiota composition and body composition?
4. The discussion touches on therapeutic potential but remains speculative. Could the authors elaborate on how these microbial signatures might inform future interventions—such as dietary modulation, probiotics, or microbiota-targeted therapies—and what safety considerations should be addressed?
5. The study references a trial registration from 2013. Clarification on how this registration aligns with the current protocol (2023–2024) would strengthen transparency and compliance with reporting standards.
Author Response
Dear Editor of Nutrients
First, my coauthors and I would like to thank you sincerely for this opportunity to cooperate. We profoundly thank the reviewers for the comments and useful suggestions to improve the paper.
This is a point-by-point list of changes made in the paper:
Reviewer 5
“This study presents a careful characterization of gut microbiota alterations in women with lipedema, integrating body composition analysis with microbial profiling to highlight the potential interplay between adipose tissue dysfunction and microbial ecology. Its thorough documentation of anthropometric parameters, DXA-derived indices, and microbial taxa provides a comprehensive overview of this underexplored condition. The pilot design and integration with existing literature underscore the novelty and complexity of the findings. However, some points merit further consideration:
1.While 16S rRNA sequencing provides useful taxonomic resolution, it does not capture functional or strain-level differences. Would the inclusion of metagenomic or metabolomic approaches enhance understanding of the therapeutic relevance and potential toxicities of microbial shifts?”
Authors thank the Reviewer for the comment. Our study was based on 16S rRNA gene sequencing, which provides robust taxonomic profiling but lacks strain-level resolution and direct insight into microbial functional potential or metabolic output. We have expanded the Limitations section as follows: “Finally, while 16S rRNA sequencing allowed robust taxonomic profiling, it does not provide strain-level resolution nor direct information on microbial functional potential or metabolic output. As a result, we cannot define which specific metabolic pathways (e.g., short-chain fatty acid, bile acid or xenobiotic metabolism) underlie the observed taxonomic shifts, nor whether these changes entail predominantly beneficial, neutral or potentially adverse effects for the host. Future studies should therefore integrate shot-gun metagenomics and targeted or untargeted metabolomics with longitudinal clinical follow-up, in order to characterise microbial gene content and circulating or faecal metabolites and thereby clarify the mechanistic, therapeutic and safety implications of microbiota alterations in lipedema.”
- “The paper suggests that intramuscular adipose tissue (IMAT) infiltration may represent a metabolically adverse phenotype in lipedema. Could further exploration and validation of this hypothesis through longitudinal imaging, biopsies, or molecular analyses strengthen the conclusions regarding disease progression and functional impairment?”
Authors thank the Reviewer for the comment. We agree that IMAT represents a metabolically adverse phenotype in lipedema, and that longitudinal imaging, muscle biopsies, and molecular analyses would strengthen the understanding of its role in disease progression and functional impairment. As our study was designed as a cross-sectional pilot analysis, such methodologies were beyond the scope of the present work. Nonetheless, we have expanded the Discussion, as follows: “To substantiate the hypothesis that IMAT represents a metabolically adverse phenotype in lipedema and a potential marker of disease progression, longitudinal and mechanistic studies will be required. Future research should combine repeated body-composition imaging (e.g., DXA or magnetic resonance imaging), histological characterization of muscle tissue and the assessment of early circulating biomarkers of metabolic stress and muscle quality, in order to validate IMAT as a biomarker of functional decline. In parallel, recent evidence has highlighted musculoskeletal ultrasound as a reliable tool for the evaluation of superficial soft tissues and lymphatic alterations in conditions such as lymphedema, providing dynamic, bedside information on tissue architecture and fluid accumulation [33,34]. The development of standardized ultra-sound protocols tailored to lipedema could facilitate more consistent assessment of subcutaneous and, where feasible, intermuscular fat compartments, including IMAT, and help integrate these measures into future diagnostic and monitoring frameworks.”
- “The correlations between specific taxa (e.g., Eggerthellaceae, Anaerostipes) and lean mass indices are intriguing but remain associative. Would the authors consider discussing potential confounders (dietary intake, physical activity, hormonal status) that may influence both microbiota composition and body composition?”
Authors thank the Reviewer for the comment. We expanded the Discussion and Limitations sections to explicitly acknowledge dietary intake, physical activity and hormonal status as potential confounders of the observed microbiota–lean mass associations, and to emphasize the purely associative nature of our findings. The following paragraphs were added to the Discussion and Limitations sections: “It is important to acknowledge that these correlations are associative and may be influenced by unmeasured confounding factors. Key determinants of both GM composition and body composition, including habitual dietary intake, physical activity levels, and hormonal status, were not quantitatively assessed in this pilot study. Variations in fiber consumption, exercise volume, or estrogen exposure can independently shape microbial metabolic profiles (e.g., SCFA production) and modulate skeletal muscle and adipose tissue physiology. Therefore, these lifestyle- and hormone-related factors may partly contribute to the microbiota–lean mass associations observed here, and should be considered in future mechanistic and longitudinal studies. […] Third, dietary intake, physical activity, and hormonal status—important determinants of both gut microbiota composition and body composition—were not quantitatively assessed, potentially introducing unmeasured confounding and partly contributing to the observed associations.”
- “The discussion touches on therapeutic potential but remains speculative. Could the authors elaborate on how these microbial signatures might inform future interventions—such as dietary modulation, probiotics, or microbiota-targeted therapies—and what safety considerations should be addressed?”
Authors thank the Reviewer for the comment. The following paragraphs were added to Discussion section, accordingly also to Reviewer 1’ comment: “Among these potential determinants, dietary patterns are particularly relevant, as they are major drivers of GM composition and function and may therefore be of interest in lipedema. High-fiber, plant-rich dietary models such as the Mediterranean diet and plant-based diets have been shown to promote SCFA-producing and anti-inflammatory taxa, increase overall microbial diversity and improve gut barrier integrity [54]. In contrast, low-carbohydrate or ketogenic diets induce distinct shifts in GM structure and bile acid metabolism, with heterogeneous effects on metabolic and inflammatory pathways [55]. Beyond macronutrient patterns, bioactive compounds with antimicrobial and immunomodulatory properties, such as capsaicin and piperine, have been shown in experimental models to reshape gut microbiota, reduce endotoxemia and low-grade inflammation, and attenuate diet-induced obesity [56]. These observations raise the hypothesis that targeted dietary strategies, including Mediterranean-style or plant-forward patterns and the controlled use of specific phytochemicals, could be explored as adjunctive approaches to modulate GM and potentially ameliorate metabolic and inflammatory features in lipedema. However, these possibilities remain speculative at present, and the use of antibiotics solely to manipulate GM cannot be recommended in a chronic condition such as lipedema, given their profound and often detrimental impact on microbial ecology. Controlled dietary intervention studies are needed to clarify whether microbiota-directed nutrition can beneficially influence disease phenotype.
At this stage, the GM signatures identified in lipedema should be regarded as exploratory biomarkers that may help design future microbiota-directed interventions rather than as therapeutic targets per se. In principle, these patterns could be used to stratify participants and define mechanistic endpoints in trials testing Mediterranean-style or fibre- and polyphenol-rich dietary patterns, as well as selected pre-, pro- or post-biotic formulations, which in other metabolic settings have been shown to favour SCFA-producing and anti-inflammatory taxa and to improve host metabolic profiles.”
- “The study references a trial registration from 2013. Clarification on how this registration aligns with the current protocol (2023–2024) would strengthen transparency and compliance with reporting standards.”
Authors thank the Reviewer for the comment. The trial registration cited (2013) refers to the original overarching study protocol under which multiple study arms were subsequently developed. The present study (2023–2024) represents one of these additional arms, designed in continuity with the initial project framework.
Each time a new study arm is created, including the one described in this manuscript, a full amendment is submitted and approved by the Institutional Ethics Committee to ensure compliance with ethical standards, updated methodology, and current regulatory requirements. Accordingly, the current protocol (2023–2024) received its own ethical approval as an amendment to the original trial. We have clarified this aspect in the revised manuscript to improve transparency and alignment with reporting standards.
We thank You for your constructive critique and we hope the review process has led to an improved manuscript.
If additional changes are warranted, we will make them.
We hope that this revised version of our manuscript may now be found suitable for publication.
Sincerely,
Rossella Cianci
Round 2
Reviewer 1 Report
Comments and Suggestions for Authors
I thank the reviewers for their diligent response and corrections done; I wish them all the best in their future endeavours
Reviewer 3 Report
Comments and Suggestions for Authors
The authors have addressed all the queries. The manuscript is now suitable for publication.